# Graph Feedback Bandits with Similar Arms

**Han Qi**[1]                **Guo Fei**[1]                **Li Zhu**[1]

[1]School of Software Engineering, Xi'an Jiaotong University, Xi'an, Shaanxi, China

## Abstract

In this paper, we study the stochastic multi-armed bandit problem with graph feedback. Motivated by the clinical trials and recommendation problem, we assume that two arms are connected if and only if they are similar (i.e., their means are close enough). We establish a regret lower bound for this novel feedback structure and introduce two UCB-based algorithms: D-UCB with problem-independent regret upper bounds and C-UCB with problem-dependent upper bounds. Leveraging the similarity structure, we also consider the scenario where the number of arms increases over time. Practical applications related to this scenario include Q&A platforms (Reddit, Stack Overflow, Quora) and product reviews in Amazon and Flipkart. Answers (product reviews) continually appear on the website, and the goal is to display the best answers (product reviews) at the top. When the means of arms are independently generated from some distribution, we provide regret upper bounds for both algorithms and discuss the sub-linearity of bounds in relation to the distribution of means. Finally, we conduct experiments to validate the theoretical results.

## 1 INTRODUCTION

The multi-armed bandit is a classical reinforcement learning problem. At each time step, the learner needs to select an arm. This will yield a reward drawn from a probability distribution (unknown to the learner). The learner's goal is to maximize the cumulative reward over a period of time steps. This problem has attracted attention from the online learning community because of its effective balance between exploration (trying out as many arms as possible) and exploitation (utilizing the arm with the best current perfor-

mance). A number of applications of multi-armed bandit can be found in online sequential decision problems, such as online recommendation systems [Li et al., 2011], online advertisement campaign [Schwartz et al., 2017] and clinical trials [Villar et al., 2015, Aziz et al., 2021].

In the standard multi-armed bandit, the learner can only observe the reward of the chosen arm. There has been existing research [Mannor and Shamir, 2011, Caron et al., 2012, Hu et al., 2020, Lykouris et al., 2020] that considers the bandit problem with side observations, wherein the learner can observe information about arms other than the selected one. This observation structure can be encoded as a graph, each node represents an arm. Node $i$ is linked to node $j$ if selecting $i$ provides information on the reward of $j$.

We study a new feedback model: if any two arms are $\epsilon$-similar, i.e., the absolute value of the difference between the means of the two arms does not exceed $\epsilon$, an edge will form between them. This means that after observing the reward of one arm, the decision-maker simultaneously knows the rewards of arms similar to it. If $\epsilon = 0$, this feedback model is the standard multi-armed bandit problem. If $\epsilon$ is greater than the maximum expected reward, this feedback model turns out to be the full information bandit problem.

As a motivating example, consider the recommendation problem in Spotify and Apple Music. After a recommender suggests a song to a user, they can observe not only whether the user liked or saved the song (reward), but also infer that the user is likely to like or save another song that is very similar. Similarity may be based on factors such as the artist, songwriter, genre, and more. As another motivating example, consider the problem of medicine clinical trials mentioned above. Each arm represents a different medical treatment plan, and these plans may have some similarities such as dosage, formulation, etc. When a researcher selects a plan, they not only observe the reward of that treatment plan but also know the effects of others similar to the selected one. The treatment effect (reward) can be either some summary information or a relative effect, such as positive or nega-

tive. Similar examples also appear in chemistry molecular simulations [Pérez et al., 2020].

In this paper, we consider two bandit models: the standard graph feedback bandit problem and the bandit problem with an increasing number of arms. The latter is a more challenging setting than the standard multi-armed bandit. Relevant applications for this scenario encompass Q&A platforms such as Reddit, Stack Overflow, and Quora, as well as product reviews on websites like Amazon and Flipkart. Answers or product reviews continuously populate these platforms means that the number of arms increases over time. The goal is to display the best answers or product reviews at the top. This problem has been previously studied and referred to as "ballooning multi-armed bandits" by Ghalme et al. [2021]. However, they require the optimal arm is more likely to arrive in the early rounds. Our contributions are as follows:

1. We propose a new feedback model, where an edge is formed when the means of two arms is less than some constant $\epsilon$. We first analyze the underlying graph $G$ of this feedback model and establish that the dominant number $\gamma(G)$ is equal to the minimum size of an independent dominating set $i(G)$, while the independent number $\alpha(G)$ is not greater than twice the dominant number $\gamma(G)$, i.e. $\gamma(G) = i(G) \geq \alpha(G)/2$. This result is helpful to the design and analysis of the following algorithms.

2. In this feedback setting, we first establish a problem-dependent regret lower bound related to $\gamma(G)$. Then, we introduce two algorithms tailored for this specific feedback structure: Double-UCB (D-UCB), utilizing twice UCB algorithms sequentially, and Conservative-UCB (C-UCB), employing a more conservative strategy during the exploration. Regret upper bounds are provided for both algorithms, with D-UCB obtaining a problem-independent bound, while C-UCB achieves a problem-dependent regret bound. Additionally, we analyze the regret bounds of UCB-N [Lykouris et al., 2020] applied to our proposed setting and prove that its regret upper bound shares the same order as D-UCB.

3. We extend D-UCB and C-UCB to the scenario where the number of arms increases over time. Our algorithm does not require the optimal arm to arrive early but assumes that the means of each arrived arm are independently sampled from some distribution, which is more realistic. We provide the regret upper bounds for D-UCB and C-UCB, along with a simple regret lower bound for D-UCB. The lower bound of D-UCB indicates that it achieves sublinear regret only when the means are drawn from a normal-like distribution, while it incurs linear regret for means drawn from a uniform distribution. In contrast, C-UCB can achieve a problem-dependent sublinear regret upper bound regardless of the means distribution.

## 2 RELATED WORKS

Bandits with side observations was first introduced by Mannor and Shamir [2011] for adversarial settings. They proposed two algorithms: ExpBan, a hybrid algorithm combining expert and bandit algorithms based on clique decomposition of the side observations graph; ELP, an extension of the well-known EXP3 algorithm [Auer et al., 2002b]. Caron et al. [2012], Hu et al. [2020] considered stochastic bandits with side observations. They proposed the UCB-N, UCB-NE, and TS-N algorithms, respectively. The regret upper bounds they obtain are of the form $\sum_{c \in \mathcal{C}} \frac{\max_{i \in c} \Delta_i \ln(T)}{(\min_{i \in c} \Delta_i)^2}$, where $\mathcal{C}$ is the clique covering of the side observation graph.

There has been some works that employ techniques beyond clique partition. Buccapatnam et al. [2014, 2018] proposed the algorithm named UCB-LP, which combine a version of eliminating arms [Even-Dar et al., 2006] suggested by Auer and Ortner [2010] with linear programming to incorporate the graph structure. This algorithm has a regret guarantee of $\sum_{i \in D} \frac{\ln(T)}{\Delta_i} + K^2$, where $D$ is a particularly selected dominating set, $K$ is the number of arms. Cohen et al. [2016] used a method based on elimination and provides the regret upper bound as $\tilde{O}(\sqrt{\alpha T})$, where $\alpha$ is the independence number of the underlying graph. Lykouris et al. [2020] utilized a hierarchical approach inspired by elimination to analyze the feedback graph, demonstrating that UCB-N and TS-N have regret bounds of order $\tilde{O}(\sum_{i \in I} \frac{1}{\Delta_i})$, where $I$ is an independent set of the graph. There is also some work that considers the case where the feedback graph is a random graph [Alon et al., 2017, Ghari and Shen, 2022, Esposito et al., 2022].

Currently, there is limited research considering scenarios where the number of arms can change. Chakrabarti et al. [2008] was the first to explore this dynamic setting. Their model assumes that each arm has a lifetime budget, after which it automatically disappears, and will be replaced by a new arm. Since the algorithm needs to continuously explore newly available arms in this setting, they only provided the upper bound of the mean regret per time step. Ghalme et al. [2021] considered the "ballooning multi-armed bandits" where the number of arms will increase but not disappear. They show that the regret grows linearly without any distributional assumptions on the arrival of the arms' means. With the assumption that the optimal arm arrives early with high probability, their proposed algorithm BL-MOSS can achieve sublinear regret. In this paper, we also only consider the "ballooning" settings but without the assumption on the optimal arm's arrival pattern. We use the feedback graph model mentioned above and assume that the means of each arrived arm are independently sampled from some distribution.

Clustering bandits Gentile et al. [2014], Li et al. [2016], Yang et al. [2022], Wang et al. [2024] are also relevant to our work. Typically, these models assume that a set of arms

(or items) can be classified into several unknown groups. Within each group, the observations associated to each of the arms follow the same distribution with the same mean. However, we do not employ a defined concept of clustering groups. Instead, we establish connections between arms by forming an edge only when their means are less than a threshold $\epsilon$, thereby creating a graph feedback structure.

## 3 PROBLEM FORMULATION

### 3.1 GRAPH FEEDBACK WITH SIMILAR ARMS

We consider a stochastic $K$-armed bandit problem with an undirected feedback graph and time horizon $T$ ($K \leq T$). At each round $t$, the learner select arm $i_t$, obtains a reward $X_t(i_t)$. Without losing generality, we assume that the rewards are bounded in $[0, 1]$ or $\frac{1}{2}$-subGaussian[1]. The expectation of $X_t(i)$ is denoted as $\mu(i) = \mathbb{E}[X_t(i)]$. Graph $G = (V, E)$ denotes the underlying graph that captures all the feedback relationships over the arms set $V$. An edge $i \leftrightarrow j$ in $E$ means that $i$ and $j$ are $\epsilon$-similarty, i.e.,

$$|\mu(i) - \mu(j)| < \epsilon,$$

where $\epsilon$ is some constant greater than 0. The learner can get a side observation of arm $i$ when pulling arm $j$, and vice versa. Let $N_i$ denote the observation set of arm $i$ consisting of $i$ and its neighbors in $G$. Let $k_t(i)$ and $O_t(i)$ denote the number of pulls and observations of arm $i$ till time $t$ respectively. We assume that each node in graph $G$ contains a self-loops, i.e., the learner will observe the reward of the pulled arm.

Let $i^*$ denote the expected reward of the optimal arm, i.e., $\mu(i^*) = \max_{i \in \{1, \dots, K\}} \mu(i)$. The gap between the expectation of the optimal arm and the suboptimal arm is denoted as $\Delta_i = \mu(i^*) - \mu(i)$. A policy, denoted as $\pi$, that selects arm $i_t$ to play at time step $t$ based on the history plays and rewards. The performance of the policy $\pi$ is measured as

$$R_T^\pi = \mathbb{E}\left[\sum_{t=1}^{T} \mu(i^*) - \mu(i_t)\right]. \tag{1}$$

### 3.2 BALLOONING BANDIT SETTING

This setting is the same as the graph feedback with similar arms above, except that the number of arms is increased over time. Let $K(t)$ denote the set of available arms at round $t$. We consider a tricky case that a new arm will arrive at each round, the total number of arms is $|K(T)| = T$. We assume that the means of each arrived arms are

---

[1]This is simply to provide a uniform description of both bounded rewards and subGaussian rewards. Our results can be easily extended to other subGaussian distributions.

independently sampled from some distribution $\mathcal{P}$. Let $a_t$ denote the arrived arm at round $t$,

$$\mu(a_1), \mu(a_2), \dots, \mu(a_T) \overset{i.i.d.}{\sim} \mathcal{P}.$$

The newly arrived arm may be connected to previous arms, depending on whether their means satisfy the $\epsilon$-similarity. In this setting, the optimal arm may vary over time. Let $i_t^*$ denote the expected reward of the optimal arm, i.e., $\mu(i_t^*) = \max_{i \in K(t)} \mu(i)$. The regret is given by

$$R_T^\pi(\mathcal{P}) = \mathbb{E}\left[\sum_{t=1}^{T} \mu(i_t^*) - \mu(i_t)\right]. \tag{2}$$

## 4 STATIONARY ENVIRONMENTS

In this section, we consider the graph feedback with similar arms in stationary environments, i.e., the number of arms remains constant. We first analyze the structure of the feedback graph. Then, we provide a problem-dependent regret lower bound. Following that, we introduce the D-UCB and C-UCB algorithm and provide the regret upper bounds.

**Dominating and Independent Dominating Sets.** A dominating set $S$ in a graph $G$ is a set of vertices such that every vertex not in $S$ is adjacent to a vertex in $S$. The domination number of $G$, denoted as $\gamma(G)$, is the smallest size of a dominating set.

An independent set contains vertices that are not adjacent to each other. An independent dominating set in $G$ is a set that both dominates and is independent. The independent domination number of $G$, denoted as $i(G)$, is the smallest size of such a set. The independence number of $G$, denoted as $\alpha(G)$, is the largest size of an independent set in $G$. For a general graph $G$, it follows immediately that $\gamma(G) \leq i(G) \leq \alpha(G)$.

**Proposition 4.1.** *Let $G$ denote the feedback graph with similar arms setting, we have $\gamma(G) = i(G) \geq \frac{\alpha(G)}{2}$.*

**Proof sketch.** The first equation can be obtained by proving that $G$ is a claw-free graph and using Lemma A.2. The second inequality can be obtained by a double counting argument. The details are in the appendix.

Proposition 4.1 shows that $\gamma(G) \leq \alpha(G) \leq 2\gamma(G)$. Once we obtain the bounds based on independence number, we also simultaneously obtain the bounds based on the domination number. Therefore, we can obtain regret bounds based on the minimum dominating set without using the feedback graph to explicitly target exploration. This cannot be guaranteed in the standard graph feedback bandit problem [Lykouris et al., 2020].

## 4.1 LOWER BOUNDS

Before presenting our algorithm, we first investigate the regret lower bound of this problem. Without loss of generality, we assume the reward distribution is $\frac{1}{2}$-subGaussian.

Let $\Delta_{min} = \mu(i^*) - \max_{j \neq i^*} \mu(j)$, $\Delta_{max} = \mu(i^*) - \min_j \mu(j)$. We assume that $\Delta_{min} < \epsilon$ in our analysis. In other words, we do not consider the easily distinguishable scenario where the optimal arm and the suboptimal arms are clearly separable. If $\Delta_{min} \geq \epsilon$, our analysis method is also applicable, but the terms related to $\Delta_{min}$ will vanish in the expressions of both the lower and upper bounds. Caron et al. [2012] has provided a lower bound of $\Omega(\log(T))$, and we present a more refined lower bound specifically for our similarity feedback.

**Theorem 4.2.** *If a policy $\pi$ is uniform good[2], for any problem instance, it holds that*

$$\liminf_{T \to \infty} \frac{R_T^\pi}{\log(T)} \geq \frac{2}{\Delta_{min}} + \frac{C_1}{\epsilon}, \qquad (3)$$

*where $C_1 = 2\log(\frac{\Delta_{max}+\epsilon}{\Delta_{max}-(\gamma(G)-2)\epsilon})$.*

*Proof.* Let $\mathcal{S}$ denote an independent dominant set that includes $i^*$. From Proposition 4.1, $|\mathcal{S}| \geq \gamma(G)$. The second-best arm is denoted as $i'$, $\mathcal{D} = \mathcal{S} \bigcup\{i'\}$. Since $\Delta_{min} < \epsilon$, $i' \notin \mathcal{S}$.

Let's construct another policy $\pi'$ for another problem-instance on $\mathcal{D}$ without side observations. If $\pi$ select arm $i_t$ at round $t$, $\pi'$ select the arm as following: if $i_t \in \mathcal{D}$, $\pi'$ select arm $i_t$ too. If $i_t \notin \mathcal{D}$, $\pi'$ will select the arm of $N_{i_t} \bigcap \mathcal{D}$ with the largest mean. Since $N_{i_t} \bigcap \mathcal{D} \neq \emptyset$, policy $\pi'$ is well-defined. It is clearly that $R_T^\pi > R_T^{\pi'}$. By the classical result of [Lai et al., 1985],

$$\liminf_{T \to \infty} \frac{R_T^{\pi'}}{\log(T)} \geq \sum_{i \in \mathcal{D}} \frac{2}{\Delta_i} = \frac{2}{\Delta_{min}} + \sum_{i \in \mathcal{S}\setminus\{i^*\}} \frac{2}{\Delta_i}. \quad (4)$$

$\forall i \in \mathcal{S}\setminus\{i^*\}, \Delta_i \in [n_i\epsilon, (n_i+1)\epsilon)$. $n_i$ is some positive integer smaller than $\frac{\Delta_{max}}{\epsilon}$. Since $\mathcal{S}$ is an independent set, $\forall i,j \in \mathcal{S}\setminus\{i^*\}, i \neq j$, we have $n_i \neq n_j$. Therefore, we can complete this proof by

$$\sum_{i \in \mathcal{S}\setminus\{i^*\}} \frac{2}{\Delta_i} \geq \sum_{j=0}^{|\mathcal{S}|-2} \frac{2}{\Delta_{max} - j\epsilon}$$

$$\geq 2\int_{-1}^{|\mathcal{S}|-2} \frac{1}{\Delta_{max} - \epsilon x} dx$$

$$\geq \frac{2\log(\frac{\Delta_{max}+\epsilon}{\Delta_{max}-(\gamma(G)-2)\epsilon})}{\epsilon}.$$

$\square$

---
[2]For more details, see [Lai et al., 1985].

---

**Algorithm 1:** D-UCB

1: **Input:** Horizon $T$, $\delta \in (0,1)$
2: Initialize $\mathcal{I} = \emptyset, \forall i, k(i) = 0, O(i) = 0$
3: **for** $t = 1$ **to** $T$ **do**
4:     **repeat**
5:        Select an arm $i_t$ that has not been observed.
6:        $\mathcal{I} = \mathcal{I} \bigcup\{i_t\}$
7:        $\forall i \in N_{i_t}$, update $k_t(i), O_t(i), \bar{\mu}_t(i)$
8:        $t = t + 1$
9:     **until** All arms have been observed at least once
10:     $j_t = \arg\max_{j \in \mathcal{I}} \bar{\mu}(j) + \sqrt{\frac{\log(\sqrt{2}T/\delta)}{O(j)}}$
11:     Select arm $i_t = \arg\max_{i \in N_{j_t}} \bar{\mu}(i) + \sqrt{\frac{\log(\sqrt{2}T/\delta)}{O(i)}}$
12:     $\forall i \in N_{i_t}$, update $k_t(i), O_t(i), \bar{\mu}_t(i)$
13: **end for**

---

Consider two simple cases. (1) $\gamma(G) = 1$. The feedback graph $G$ is a complete graph or some graph with independence number less than 2. Then $C_1 = 0$, the lower bound holds strictly when $G$ is not a complete graph. (2) $\gamma(G) = \frac{\Delta_{max}}{\epsilon}$, $C_1 = 2\log(\frac{1}{2}\gamma(G) + \frac{1}{2})$. In this case, the terms in the lower bound involving $\epsilon$ have the same order $O(\log(\gamma(G)))$ as the corresponding terms in the upper bound of the following proposed algorithms.

## 4.2 DOUBLE-UCB

This particular feedback structure inspires us to distinguish arms within the independent set first. This is a straightforward task because the distance between the mean of each arm in the independent set is greater than $\epsilon$. Subsequently, we learn from the arm with the maximum confidence bound in the independent set and its neighborhood, which may include the optimal arm. Our algorithm alternates between the two processes simultaneously.

Algorithm 1 shows the pseudocode of our method. Steps 4-9 identify an independent set $\mathcal{I}$ in $G$, play each arm in the independent set once. This process does not require knowledge of the complete graph structure and requires at most $\alpha(G)$ rounds. Step 10 calculates the arm $j_t$ with the maximum upper confidence bound in the independent set. After a finite number of rounds, the optimal arm is likely to fall within $N_{j_t}$. Step 11 use the UCB algorithm again to select arm in $N_{j_t}$. This algorithm employs UCB rules twice for arm selection, hence it is named Double-UCB.

### 4.2.1 Regret Analysis of Double-UCB

We use $\mathcal{I}(N_i)$ to denote the set of all independent dominating sets of graph formed by $N_i$. Let

$$\mathcal{I}(i^*) = \bigcup_{i \in N_{i^*}} \mathcal{I}(N_i).$$

Note that, the elements in $\mathcal{I}(i^*)$ are independent sets rather than individual nodes, and $\forall I \in \mathcal{I}(i^*), |I| \le 2$.

**Theorem 4.3.** *Assume that the reward distribution is $\frac{1}{2}$-subgaussian or bounded in $[0,1]$, set $\delta = \frac{1}{T}$, the regret of Double-UCB after $T$ rounds is upper bounded by*

$$R_T^\pi \le 32(\log(\sqrt{2}T))^2 \max_{I \in \mathcal{I}(i^*)} \sum_{i \in I \setminus \{i^*\}} \frac{1}{\Delta_i} + C_2 \frac{\log(\sqrt{2}T)}{\epsilon}$$
$$+ \Delta_{max} + 4\epsilon + 1,$$

(5)

*where $C_2 = 8(\log(2\gamma(G)) + \frac{\pi^2}{3})$.*

*Recall that we assume $\Delta_{min} < \epsilon$, then $\forall i \in N_{i^*}, |N_i| > 1$. The index set $\{i \in I \setminus \{i^*\}\}$ in the summation of the first term is non-empty. If $\Delta_{min} \ge \epsilon$, $\mathcal{I}(i^*) = \{i^*\}$. The first term will vanish.*

**Proof sketch.** The regret can be analyzed in two parts. The first part arises from Step 9 selecting $j_t$ whose neighborhood does not contain the optimal arm. The algorithm can easily distinguish the suboptimal arms in $N_{j_t}$ from the optimal arm. The regret caused by this part is $O(\frac{\log(T)}{\epsilon})$. The second part of the regret comes from selecting a suboptimal arm in $N_{j_t}, i^* \in N_{j_t}$. This part can be viewed as applying UCB rule on a graph with an independence number less than 2.

Since $\forall I \in \mathcal{I}(i^*), |I| \le 2$, we have

$$\max_{I \in \mathcal{I}(i^*)} \sum_{i \in I \setminus \{i^*\}} \frac{1}{\Delta_i} \le \frac{2}{\Delta_{min}}.$$

Therefore, the regret upper bound can be denoted as

$$R_T^\pi \le \frac{64(\log(\sqrt{2}T))^2}{\Delta_{min}} + C_2 \frac{\log(\sqrt{2}T)}{\epsilon} + \Delta_{max} + 4\epsilon + 1.$$

(6)

Our upper bound suffers an extra logarithm term compared to the lower bound Theorem 4.2. The inclusion of this additional logarithm appears to be essential if one uses a gap-based analysis similar to [Cohen et al., 2016]. This issue has been discussed in [Lykouris et al., 2020].

From Theorem 4.3, we have the following gap-free upper bound

**Corollary 4.4.** *The regret of Double-UCB is bounded by $16\sqrt{T}\log(\sqrt{2}T) + C_2 \frac{\log(\sqrt{2}T)}{\epsilon} + \Delta_{max} + 4\epsilon + 1$.*

### 4.3 CONSERVATIVE UCB

Double-UCB is a very natural algorithm for similar arms setting. For this particular feedback structure, we propose the conservative UCB, which simply modifies Step 10 of Algorithm 1 to $i_t = \arg\max_{i \in N_{j_t}} \bar{\mu}(i) - \sqrt{\frac{\log(\sqrt{2}T/\delta)}{O(i)}}$. This form of the index function is conservative when exploring arms in $N_{j_t}$. It does not immediately try each arm but selects those that have been observed a sufficient number of times. Algorithm 2 shows the pseudocode of C-UCB.

---

**Algorithm 2:** C-UCB

1: **Input:** Horizon $T$, $\delta \in (0,1)$
2: Initialize $\mathcal{I} = \emptyset, \forall i, k(i) = 0, O(i) = 0$
3: **for** $t = 1$ to $T$ **do**
4:     Steps 4-10 in D-UCB
5:     Select arm $i_t = \arg\max_{i \in N_{j_t}} \bar{\mu}(i) - \sqrt{\frac{\log(\sqrt{2}T/\delta)}{O(i)}}$
6:     $\forall i \in N_{i_t}$, update $k_t(i), O_t(i), \bar{\mu}_t(i)$
7: **end for**

---

#### 4.3.1 Regret Analysis of Conservative UCB

Let $G_{2\epsilon}$ denote the subgraph with arms $\{i \in V : \mu(i^*) - \mu(i) < 2\epsilon\}$. The set of all independent dominating sets of graph $G_{2\epsilon}$ is denoted as $\mathcal{I}(G_{2\epsilon})$. We can also define $\mathcal{I}(G_\epsilon)$ in this way. Define $\Delta_{2\epsilon}^{min} = \min_{i,j \in G_{2\epsilon}} \{|\mu(i) - \mu(j)|\}$.

We divide the regret into two parts. The first part is the regret caused by choosing arm in $N_{j_t}, i^* \notin N_{j_t}$, and the analysis for this part follows the same approach as D-UCB.

The second part is the regret of choosing the suboptimal arms in $N_{j_t}, i^* \in N_{j_t}$. It can be proven that if the optimal arm is observed more than $\frac{4\log(\sqrt{2}T/\delta)}{(\Delta_{2\epsilon}^{min})^2}$ times, the algorithm will select the optimal arm with high probability. Intuitively, for any arm $i \in N_{j_t}, i \ne i^*$, the following events hold with high probability:

$$\bar{\mu}(i^*) - \sqrt{\frac{\log(\sqrt{2}T/\delta)}{O(i^*)}} > \mu(i^*) - \Delta(i) = \mu(i), \quad (7)$$

and

$$\bar{\mu}(i) - \sqrt{\frac{\log(\sqrt{2}T/\delta)}{O(i)}} < \mu(i). \quad (8)$$

Since the optimal arm satisfies Equation (7) and the suboptimal arms satisfy Equation (8) with high probability, the suboptimal arms are unlikely to be selected.

The key to the problem lies in ensuring that the optimal arm can be observed $\frac{4\log(\sqrt{2}T/\delta)}{(\Delta_{2\epsilon}^{min})^2}$ times. Since in the graph formed by $N_{j_t}$, all arms are connected to $j_t$. As long as the time steps are sufficiently long, arm $j_t$ will inevitably be observed more than $\frac{4\log(\sqrt{2}T/\delta)}{(\Delta_{2\epsilon}^{min})^2}$ times, then the arms with means less than $\mu(j_t)$ will be ignored (similar to Equation (7) and Equation (8)). Choosing arms that the means between $(\mu(j_t), \mu(i^*))$ will always observe the optimal arm, so the optimal arm can be observed $\frac{4\log(\sqrt{2}T/\delta)}{(\Delta_{2\epsilon}^{min})^2}$ times. Therefore, we have the following theorem:

**Theorem 4.5.** *Under the same conditions as Theorem 4.3, the regret of C-UCB is upper bounded by*

$$R_T^\pi \le \frac{32\epsilon \log(\sqrt{2}T)}{(\Delta_{2\epsilon}^{min})^2} + C_2 \frac{\log(\sqrt{2}T)}{\epsilon} + \Delta_{max} + 2\epsilon \quad (9)$$

The regret upper bound of C-UCB decreases by a logarithmic factor compared to D-UCB, but with an additional problem-dependent term $\Delta_{2\epsilon}$. This upper bound still does not match the lower bound in Theorem 4.2, as $\Delta_{2\epsilon}^{min} \leq \Delta_{min}$. If we ignore the terms independent of $T$, the regret upper bound of C-UCB matches the lower bound of $\Omega(\log(T))$.

### 4.4 UCB-N

UCB-N has been analyzed in the standard graph feedback model [Caron et al., 2012, Hu et al., 2020, Lykouris et al., 2020]. One may wonder whether UCB-N can achieve similar regret upper bounds. In fact, if UCB-N uses the same upper confidence function as ours, UCB-N has a similar regret upper bound to D-UCB. We have the following theorem:

**Theorem 4.6.** *Under the same conditions as Theorem 4.3, the regret of UCB-N is upper bounded by*

$$R_T^\pi \leq \frac{32(\log(\sqrt{2}T))^2}{\Delta_{min}} + C_3 \frac{\log(\sqrt{2}T)}{\epsilon} + \Delta_{max} + 2\epsilon + 1,$$
(10)

*where $C_3 = 8(\log(2\gamma(G)) + \frac{\pi^2}{6})$.*

**Remark 4.7.** *The gap-free upper bound of UCB-N is also $O(\sqrt{T}\log(T))$. Due to the similarity assumption, the regret upper bound of UCB-N is improved compared to [Lykouris et al., 2020]. D-UCB and C-UCB are specifically designed for similarity feedback structures and may fail in the case of standard graph feedback settings. This is because the optimal arm may be connected to an arm with a very small mean, so the $j_t$ selected in Step 9 may not necessarily include the optimal arm. However, under the ballooning setting, UCB-N cannot achieve sublinear regret. D-UCB and C-UCB can be naturally applied in this setting and achieve sublinear regret under certain conditions.*

## 5 BALLOONING ENVIRONMENTS

This section considers the setting where arms are increased over time. This problem is highly challenging, as prior research relied on strong assumptions to achieve sublinear regret. The graph feedback structure we propose is particularly effective for this setting. Intuitively, if an arrived arm has a mean very close to arms that have already been distinguished, the algorithm does not need to distinguish it further. This may lead to a significantly smaller number of truly effective arrived arms than $T$, making it easier to obtain a sublinear regret bound.

### 5.1 DOUBLE-UCB FOR BALLOONING SETTING

Algorithm 3 shows the pseudocode of our method Double-UCB-BL. For any set $\mathcal{S}$, let $N_{\mathcal{S}}$ denote the set of arms linked

---

**Algorithm 3:** Double-UCB for Ballooning Setting

1: **Input:** Horizon $T$, $\delta \in (0, 1)$
2: Initialize $\mathcal{I} = \emptyset, \forall i, k(i) = 0, O(i) = 0$
3: **for** $t = 1$ **to** $T$ **do**
4:     Arm $a_t$ arrives
5:     Feedback graph $G$ is updated
6:     **if** $a_t \notin N_{\mathcal{I}}$ **then**
7:         $\mathcal{I} = \mathcal{I} \bigcup \{a_t\}$
8:     **end if**
9:     $j_t = \arg\max_{j \in \mathcal{I}} \bar{\mu}(j) + \sqrt{\frac{\log(\sqrt{2}T/\delta)}{O(j)}}$
10:     Pulls arm $i_t = \arg\max_{i \in N_{j_t}} \bar{\mu}(i) + \sqrt{\frac{\log(\sqrt{2}T/\delta)}{O(i)}}$
11:     $\forall i \in N_{i_t}$, update $k_t(i), O_t(i), \bar{\mu}_t(i)$
12: **end for**

---

to $\mathcal{S}$, i.e., $N_{\mathcal{S}} = \bigcup_{i \in \mathcal{S}} N_i$. Upon the arrival of each arm, first check whether it is in $N_{\mathcal{I}}$. If it is not, add it to $\mathcal{I}$ to form a new independent set. The construction of the independent set $\mathcal{I}$ is formed online as arms arrive, while the other parts remain entirely consistent with Double-UCB.

#### 5.1.1 Regret Analysis

Let $\mathcal{I}_t$ denote the independent set at time $t$ and $\alpha_t^* \in \mathcal{I}_t$ denote the arm that include the optimal arm $i_t^*$. Define $\mathcal{A} = \{a_t : t \in [T], \mu(a_t) \in N_{\alpha_t^*}\}$. To simplify the problem, we only consider the problem instances that satisfy the following assumption:

**Assumption 5.1.** $\forall i \neq j, \Delta_{min}^T \leq |\mu(i) - \mu(j)| \leq \Delta_{max}^T$, $\Delta_{min}^T, \Delta_{max}^T$ is some constant.

The first challenge in ballooning settings is the potential existence of many arms with means very close to the optimal arm. That is, the set $\mathcal{A}$ may be very large. We first define a quantity that is easy to analyze as the expected upper bound for all arms falling into $N_{\alpha_t^*}$. Define

$$M = \mathbb{E}[\sum_{t=1}^T \mathbb{1}\{|\mu(a_t) - \mu(i_t^*)| < 2\epsilon\}]$$
$$= \sum_{t=1}^T \mathbb{P}(|\mu(a_t) - \mu(i_t^*)| < 2\epsilon)$$
(11)

It's easy to verify that $\{\mu(a_t) \in N_{\alpha_t^*}\} \subseteq \{|\mu(a_t) - \mu(i_t^*)| < 2\epsilon\}$. We have $\mathbb{E}[|\mathcal{A}|] \leq M$.

The second challenge is that our regret is likely related to the independence number, yet under the ballooning setting, the graph's independence number is a random variable. Denote the independence number as $\alpha(G_T^{\mathcal{P}})$. We attempt to address this issue by providing a high-probability upper bound for the independence number. Let $X, Y \overset{i.i.d.}{\sim} \mathcal{P}$, then

$$p = \mathbb{P}(|X - Y| \leq \epsilon) = \int_{-\epsilon}^{\epsilon} f_{X-Y}(z)dz, \quad (12)$$

where $f_{X-Y}$ is the probability density function of $X-Y$. Let $b = \frac{1}{1-p}$, Lemma A.3 has proved a high-probability upper bound of $\alpha(G_T^{\mathcal{P}})$ related to $b$.

Now, we can give the following upper bound of Double-UCB.

**Theorem 5.2.** *Assume that the reward distribution is $\frac{1}{2}$-subgaussian or bounded in $[0,1]$, set $\delta = \frac{1}{T}$, the regret of Double-UCB after $T$ rounds is upper bounded by*

$$R_T^\pi \leq 40 \max\{\log_b T, 1\}\Delta_{max}^T \frac{\log(\sqrt{2}T)}{\epsilon^2} + 2\Delta_{max}^T$$
$$+ 4\sqrt{6TM\log(\sqrt{2}T)} + 2\epsilon + 2T\epsilon e^{-M}, \tag{13}$$

If $\mathcal{P}$ is Gaussian distribution, we have the following corollary:

**Corollary 5.3.** *If $\mathcal{P}$ is the Gaussian distribution $\mathcal{N}(0,1)$, we have $M = O(\log(T)e^{2\epsilon\sqrt{2\log(T)}})$ and $M = \Omega(\log(T)\sqrt{\log(T)})$. The asymptotic regret upper bound is of order*

$$O\left(\log(T)\sqrt{Te^{2\epsilon\sqrt{2\log(T)}}}\right).$$

*The order of $e^{\sqrt{2\log(T)}}$ is smaller than any power of $T$. For example, if $T > e^n$, $n$ is a positive integer, we have $e^{\sqrt{2\log(T)}} \leq T^{\sqrt{2/n}}$.*

**Lower Bounds of Double-UCB** Define $\mathcal{B} = \{a_t : t \in [T], \frac{\epsilon}{2} < \mu(i_t^*) - \mu(a_t) < \epsilon\}$. Then we have $\mathcal{B} \subset \mathcal{A}$. Let

$$B = \mathbb{E}\left[\sum_{t=1}^T \mathbb{1}\{\frac{\epsilon}{2} < \mu(i_t^*) - \mu(a_t) < \epsilon\}\right].$$

We have $\mathbb{E}[|\mathcal{B}|] = B$. If arm $a_t$ arrives and falls into set $\mathcal{B}$ at round $t$, then the arm will be selected at least once, unless the algorithm select $j_t \neq \alpha_t^*$ in Step 9. Note that if an arm in $\mathcal{B}$ is selected at some round, the resulting regret is at least $\frac{\epsilon}{2}$. Once we estimate the size of $|\mathcal{B}|$ and the number of rounds with $j_t \neq \alpha_t^*$, a simple regret lower bound can be obtained. We have the following lower bound:

**Theorem 5.4.** *Assume that the reward distribution is $\frac{1}{2}$-subgaussian or bounded in $[0,1]$, set $\delta = \frac{1}{T}$, the regret lower bound of Double-UCB is*

$$R_T^\pi \geq \frac{B\epsilon}{4}(1 - e^{-B/8}) - 20 \max\{\log_b T, 1\}\frac{\log(\sqrt{2}T)}{\epsilon} - \epsilon \tag{14}$$

If $\mathcal{P}$ is the uniform distribution $U(0,1)$, we can calculate that $B \geq \frac{(1-\epsilon)\epsilon}{2}T$. If $\mathcal{P}$ is the half-triangle distribution with probability density function as $f(x) = 2(1-x)\mathbb{1}\{0 < x < 1\}$. We can also calculate $B \geq \frac{3\epsilon^2(1-\epsilon)^2}{4}T$. Therefore, their regret lower bounds are of linear order.

---

**Algorithm 4:** C-UCB for Ballooning Setting

---

1: **Input:** Horizon $T$, $\delta \in (0,1)$
2: Initialize $\mathcal{I} = \emptyset, \forall i, k(i) = 0, O(i) = 0$
3: **for** $t = 1$ **to** $T$ **do**
4:     Steps 4-9 in Double-UCB-BL
5:     Pulls arm $i_t = \arg\max_{i \in N_{j_t}} \bar{\mu}(i) - \sqrt{\frac{\log(\sqrt{2}T/\delta)}{O(i)}}$
6:     $\forall i \in N_{i_t}$, update $k_t(i), O_t(i), \bar{\mu}_t(i)$
7: **end for**

---

## 5.2 C-UCB FOR BALLOONING SETTING

The failure on the common uniform distribution limits the D-UCB's application. The fundamental reason is the aggressive exploration strategy of the UCB algorithm, which tries to explore every arm that enters set $\mathcal{A}$ as much as possible. In this section, we apply C-UCB to ballooning settings, which imposes no requirements on the distribution $\mathcal{P}$.

Algorithm 4 shows the pseudocode of conservative UCB (C-UCB). This algorithm is almost identical to D-UCB, with the only change being that the rule for selecting an arm is $\arg\max_{i \in N_{j_t}} \bar{\mu}(i) - \sqrt{\frac{\log(\sqrt{2}T/\delta)}{O(i)}}$. This improvement avoids exploring every arm that enters $N_{j_t}$. It is only chosen if it has been observed a sufficient number of times and its confidence lower bound is the largest. We have the following regret upper bound for C-UCB:

**Theorem 5.5.** *Assume that the reward distribution is $\frac{1}{2}$-subgaussian or bounded in $[0,1]$, set $\delta = \frac{1}{T}$, the regret of C-UCB is upper bounded by*

$$R_T^\pi \leq 40 \max\{\log_b T, 1\}\Delta_{max}^T \frac{\log(\sqrt{2}T)}{\epsilon^2} + 2\Delta_{max}^T$$
$$+ \frac{96\epsilon(\log(eT))^2}{(\Delta_{min}^T)^2} + 4\epsilon \tag{15}$$

The arms arrive one by one in ballooning setting, and the optimal arm may change over time. Therefore, the regret upper bound depends on $\Delta_{min}^T$ rather than $\Delta_{2\epsilon}$. Compared to Theorem 5.2, the upper bound for C-UCB does not involve $M$, making it independent of the distribution $\mathcal{P}$. Under the conditions of Assumption 5.1, it achieves a regret upper bound of $O((\log(T))^2)$.

# 6 EXPERIMENTS

## 6.1 STATIONARY SETTINGS

We first compared the performance of UCB-N under standard graph feedback and graph feedback with similar arms

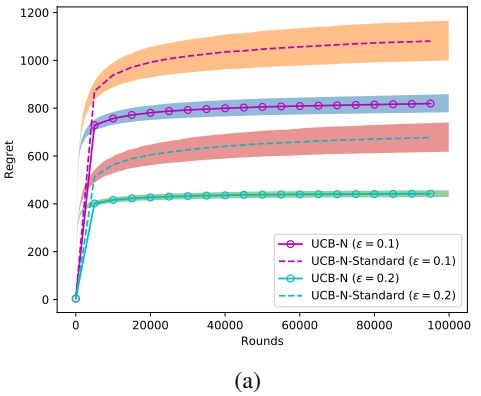
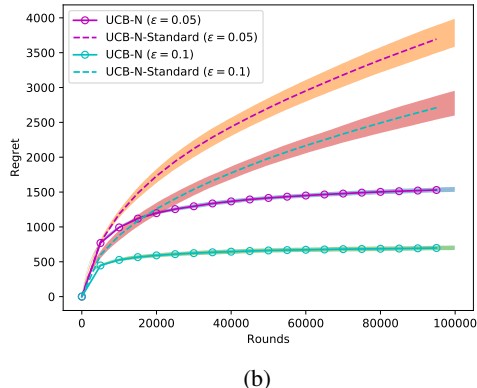

(a)                                                      (b)

Figure 1: "UCB-N ($\epsilon = 0.1$)": Graph feedback **with** similarity structure. "UCB-N-Standard ($\epsilon = 0.1$)": Graph feedback **without** similarity structure, but the graph used has roughly the same independence number with the former setting. Settings with $T = 10^5, K = 10^4$. (a) $\epsilon = 0.1, 0.2$ for Gaussian rewards. (b) $\epsilon = 0.05, 0.1$ for Bernoulli rewards.

[3]. The purpose of this experiment is to show that the similarity structure improves the performance of the original UCB algorithm. To ensure fairness, the problem instances we use in both cases have roughly the same independence number. In the standard graph feedback, we also use a random graph, generating edges with a probability calculated by Equation (12). The graph generated in this way has roughly the same independence number as the graph in the $\epsilon$-similarity setting. In particular, if $\mathcal{P}$ is the Gaussian distribution $\mathcal{N}(0, 1)$, then

$$p = \sqrt{2}(2\Phi(\frac{\epsilon}{\sqrt{2}}) - 1).$$

If $\mathcal{P}$ is the uniform distribution $U(0, 1)$,

$$p = 1 - (1 - \epsilon)^2.$$

For each value of $\epsilon$, we generate 50 different problem instances. The expected regret is averaged on the 50 instances. The 95% confidence interval is shown as a semi-transparent region in the figure. Figure 1 shows the performance of UCB-N under Gaussian rewards. It can be observed that the regret of UCB-N in our settings is smaller than standard graph feedback, thanks to the similarity structure. Additionally, the regret decreases as $\epsilon$ increases, consistent with theoretical results.

We then compared the performance of UCB-N, D-UCB, and C-UCB algorithms. Figure 2 shows the performance of the three algorithms with Gaussian and Bernoulli rewards. Although D-UCB and UCB-N have similar regret bounds, the experimental performance of D-UCB and C-UCB is better than UCB-N. This may be because D-UCB and C-UCB directly learn on an independent set, effectively leveraging the graph structure features of similar arms.

[3]Our code is available at https://github.com/qh1874/GraphBandits_SimilarArms

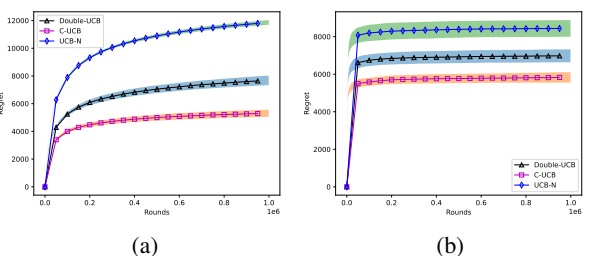

(a)                              (b)

Figure 2: Settings with $T = 10^6, K = 10^5, \epsilon = 0.01$. Bernoulli rewards (a), Gaussian rewards (b).

## 6.2 BALLOONING SETTINGS

UCB-N is not suitable for ballooning settings since it would select each arrived arm at least once. The BL-Moss algorithm [Ghalme et al., 2021] is specifically designed for the ballooning setting. However, this algorithm assumes that the optimal arm is more likely to appear in the early rounds and requires prior knowledge of the parameter $\lambda$ to characterize this likelihood, which is not consistent with our setting. We only compare D-UCB and C-UCB with different distributions $\mathcal{P}$.

Figure 3 show the experimental results of ballooning settings. When $\mathcal{P}$ follows a standard normal distribution, D-UCB and C-UCB exhibit similar performance. However, when $\mathcal{P}$ is a uniform distribution $U(0, 1)$ or half-triangle distribution with distribution function as $1 - (1 - x)^2$, D-UCB fails to achieve sublinear regret, while C-UCB still performs well. These results are consistent across different values of $\epsilon$.

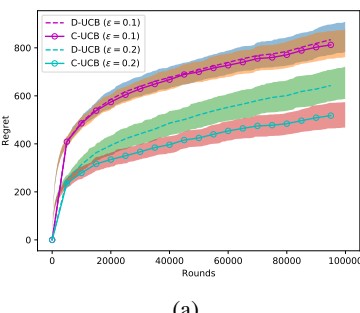 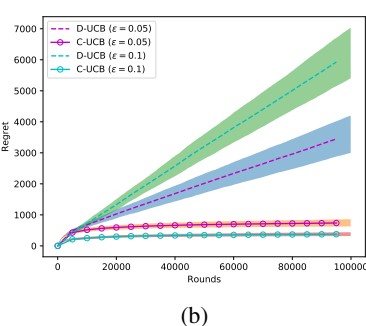 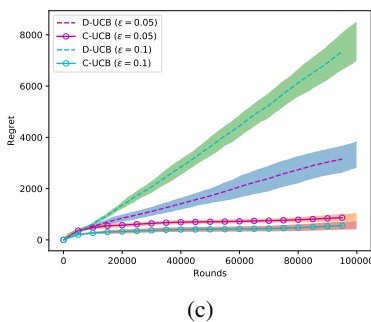

| (a) | (b) | (c) |

Figure 3: Ballooning Setting. (a) Gaussian arms with $\mathcal{P}$ as $\mathcal{N}(0,1)$ and $\epsilon = 0.1, 0.2$. (b) Bernoulli arms with $\mathcal{P}$ as $U(0,1)$ and $\epsilon = 0.05, 0.1$. (c) Bernoulli arms with $\mathcal{P}$ being the half-triangle distribution and $\epsilon = 0.05, 0.1$.

## 7 CONCLUSION

In this paper, we have introduced a new graph feedback bandit model with similar arms. For this model, we proposed two different UCB-based algorithms (D-UCB, C-UCB) and provided regret upper bounds. We then extended these two algorithms to the ballooning setting. In this setting, the application of C-UCB is more extensive than D-UCB. D-UCB can only achieve sublinear regret when the mean distribution is Gaussian, while C-UCB can achieve problem-dependent sublinear regret regardless of the mean distribution.

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

In this appendix, we provide the detailed proof of theorems and corollaries in the main text. Specifically, we provide detailed proofs of Proposition 4.1, Theorem 4.3, Theorem 4.6, Theorem 5.2, Corollary 5.3 and Theorem 5.5. The proof of Theorem 4.5 can be easily obtained from the analysis of Theorem 5.5. The proof of Theorem 5.4 is similar to that of Theorem 5.2. We omit their proofs. Beforehand, we give some well-known results to simplify the proof.

## A FACTS AND LEMMAS

**Lemma A.1.** *Assume that $X_i$ are independent random variables. $\mu = \mathbb{E}[X_i], \bar{\mu} = \frac{1}{n}\sum_{i=1}^{n} X_i$. If $X_i$ is bounded in $[0, 1]$ or $\frac{1}{2}$-subGaussian, for any $\delta \in (0, 1)$, with probability as least $1 - \frac{\delta^2}{T^2}$,*

$$|\bar{\mu} - \mu| \leq \sqrt{\frac{\log(\sqrt{2}T/\delta)}{n}}.$$

**Lemma A.2.** *Allan and Laskar [1978] If $G$ is a claw-free graph, then $\gamma(G) = i(G)$.*

**Lemma A.3.** *Assume that $\mu(a_t) \overset{i.i.d.}{\sim} \mathcal{P}$. Let $G_T^{\mathcal{P}}$ denote the graph constructed by $\mu(a_1), \mu(a_2), ..., \mu(a_T), \alpha(G_T^{\mathcal{P}})$ is the independent number of $G_T^{\mathcal{P}}$. Then*

$$\mathbb{P}(\alpha(G_T^{\mathcal{P}}) \geq 5\max\{\log_b T, 1\}) \leq \frac{1}{T^5}, \tag{16}$$

*where $b$ is some constant related to $\mathcal{P}$.*

*Proof.* Let $X, Y \overset{i.i.d.}{\sim} \mathcal{P}$, then

$$\mathbb{P}(|X - Y| \leq \epsilon) = \int_{-\epsilon}^{\epsilon} f_{X-Y}(z)dz = p,$$

where $f_{X-Y}$ is the probability density function of $X - Y$. This means that in $G_T^{\mathcal{P}}$, the probability of any two nodes being connected by an edge is $p$. Hence, $G_T^{\mathcal{P}}$ is a random graph.

Let $Z_k$ be the number of independent sets of order $k$. Let $b = \frac{1}{1-p}$, $k = \lceil 5\log_b T \rceil$,

$$\begin{aligned}
\mathbb{P}(\alpha(G_T^{\mathcal{P}}) \geq 5\log_b T) &\leq \mathbb{P}(Z_k \geq 1) \\
&\leq \mathbb{E}[Z_k] \\
&= \binom{T}{k}(1-p)^{\binom{k}{2}} \\
&\overset{(a)}{\leq} \left(\frac{Te}{k\sqrt{1-p}}(1-p)^{k/2}\right)^k \\
&\leq \left(\frac{e\sqrt{b}}{k}\right)^k\left(\frac{1}{T^{1.5}}\right)^k,
\end{aligned} \tag{17}$$

where $(a)$ uses the fact that $\binom{T}{k} \leq \left(\frac{Te}{k}\right)^k$.

If $b < T, k \geq 5 > e$. We have,

$$\mathbb{P}(\alpha(G_T^{\mathcal{P}}) \geq 5\log_b T) \leq \left(\frac{e\sqrt{b}}{k}\right)^k\left(\frac{1}{T^{1.5}}\right)^k = \left(\frac{e\sqrt{b}}{k\sqrt{T}}\right)^k\left(\frac{1}{T}\right)^k \leq \frac{1}{T^5}.$$

If $b \geq T$, i.e., $1 - p \leq \frac{1}{T}$, we have

$$\mathbb{P}(\alpha(G_T^{\mathcal{P}}) \geq 5) \leq \mathbb{P}(Z_5 \geq 1) \leq \binom{T}{5}(1-p)^{\binom{5}{2}} \leq \frac{1}{T^5}.$$

Therefore,

$$\mathbb{P}(\alpha(G_T^{\mathcal{P}}) \geq 5\max\{\log_b T, 1\}) \leq \frac{1}{T^5}.$$

$\square$

**Lemma A.4** (Chernoff Bounds). *Let $X_i$ be independent Bernoulli random variable. Let $X$ denote their sum and let $\mu = \mathbb{E}[X]$ denote the sum's expected value.*

$$\mathbb{P}(X \geq (1+\delta)\mu) \leq e^{-\frac{\delta^2 \mu}{2+\delta}}, \delta \geq 0,$$

$$\mathbb{P}(X \leq (1-\delta)\mu) \leq e^{-\frac{\delta^2 \mu}{2}}, 0 < \delta < 1.$$

**Lemma A.5.** *Abramowitz et al. [1988] For a Gaussian distributed random variable $X$ with mean $\mu$ and variance $\sigma^2$, for any $a > 0$,*

$$\frac{1}{\sqrt{2\pi}} \frac{a}{1+a^2} e^{-\frac{a^2}{2}} \leq \mathbb{P}(X - \mu > a\sigma) \leq \frac{1}{a + \sqrt{a^2 + 4}} e^{-\frac{a^2}{2}}.$$

# B  PROOFS OF PROPOSITION 4.1

(1) We first prove $\gamma(G) = i(G)$. A claw-free graph is a graph that does not have a claw as an induced subgraph or contains no induced subgraph isomorphic to $K_{1,3}$. From Lemma A.2, we just need to prove $G$ is claw-free.

Assuming $G$ has a claw, meaning there exists nodes $a, b, c, d$, such that $a$ is connected to $b, c, d$, while $b, c, d$ are mutually unconnected. Consider nodes $b, c$. $b$ and $c$ must have a mean greater than $a$, and the other must have a mean smaller than $a$. Otherwise, the mean difference between $b$ and $c$ will be smaller than $\epsilon$, and an edge will form between them. Since $d$ is connected to $a$, this would lead to an edge between $d$ and $b$ or $d$ and $c$. This is a contradiction. Therefore, $G$ is claw-free.

(2) $\alpha(G) \leq 2i(G)$. Let $I^*$ be a maximum independent set and $I$ be a minimum independent dominating set. Note that any vertex of $I$ is adjacent (including the vertex itself in the neighborhood) to at most two vertices in $I^*$, and that each vertex of $I^*$ is adjacent to at least one vertex of $I$. So by a double counting argument, when counting once the vertices of $I^*$, we can choose one adjacent vertex in $I$, and we will have counted at most twice the vertices of $I$.

# C  PROOFS OF THEOREM 4.3

Let $\mathcal{I}$ denote the independent set obtained after running Step 4-8 in Algorithm 1. The obtained $\mathcal{I}$ may vary with each run. We first fix $\mathcal{I}$ for analysis and then take the supremum of the results with respect to $\mathcal{I}$, obtaining an upper bound independent of $\mathcal{I}$.

Let $\alpha^* \in \mathcal{I}$ denotes the arm that includes the optimal arm, i.e., $i^* \in N_{\alpha^*}$. Let $\mathcal{I} = \{\alpha_1, \alpha_2, ..., \alpha^*, ..., \alpha_{|\mathcal{I}|}\}$. The regret can be divided into two parts: one part is the selection of arms $i \notin N_{\alpha^*}$ and the other part is the selection of arms $i \in N_{\alpha^*}$:

$$\sum_{t=1}^{T} \sum_{i \in V} \Delta_i \mathbb{1}\{i_t = i\} = \sum_{t=1}^{T} \sum_{i \notin N_{\alpha^*}} \Delta_i \mathbb{1}\{i_t = i\} + \sum_{t=1}^{T} \sum_{i \in N_{\alpha^*}} \Delta_i \mathbb{1}\{i_t = i\} \tag{18}$$

We first focus on the expected regret incurred by the first part. Let $\Delta'_{\alpha_j} = \mu(\alpha^*) - \mu(\alpha_j)$, $j_t \in \mathcal{I}$ denote the arm linked to the selected arm $i_t$(Step 9 in Algorithm 1).

$$\sum_{t=1}^{T} \sum_{i \notin N_{\alpha^*}} \Delta_i \mathbb{1}\{i_t = i\} = \sum_{j=1}^{|\mathcal{I}|} \sum_{t=1}^{T} \sum_{i \in N_{\alpha_j}} \Delta_i \mathbb{1}\{i_t = i\} \leq \sum_{j=1}^{|\mathcal{I}|} (\Delta'_{\alpha_j} + 2\epsilon) \sum_{t=1}^{T} \mathbb{1}\{j_t = \alpha_j\}. \tag{19}$$

The last inequality uses the following two facts:

$$\Delta_i = \mu(i^*) - \mu(i) = \mu(i^*) - \mu(\alpha^*) + \mu(\alpha^*) - \mu(\alpha_j) + \mu(\alpha_j) - \mu(i) \leq \Delta'_{\alpha_j} + 2\epsilon,$$

and

$$\sum_{t=1}^{T} \sum_{i \in N_{\alpha_j}} \mathbb{1}\{i_t = i\} = \sum_{t=1}^{T} \mathbb{1}\{j_t = \alpha_j\}.$$

Recall that $O_t(i)$ denotes the number of observations of arm $i$ till time $t$. Let $c_t(i) = \sqrt{\frac{2 \log(T^2/\delta)}{O_t(i)}}$, $\bar{X}_s(i)$ denote average reward of arm $i$ after observed $s$ times, $c_s(i) = \sqrt{\frac{2 \log(T^2/\delta)}{s}}$ . For any $\alpha_j \in \mathcal{I}$,

$$\sum_{t=1}^{T} \mathbb{1}\{j_t = \alpha_j\} \leq \ell_{\alpha_j} + \sum_{t=1}^{T} \mathbb{1}\{j_t = \alpha_j, O_t(\alpha_j) \geq \ell_{\alpha_j}\}$$

$$\leq \ell_{\alpha_j} + \sum_{t=1}^{T} \mathbb{1}\{\bar{\mu}_t(\alpha_j) + c_t(\alpha_j) \geq \bar{\mu}_t(\alpha^*) + c_t(\alpha^*), O_t(\alpha_j) \geq \ell_{\alpha_j}\}$$

$$\leq \ell_{\alpha_j} + \sum_{t=1}^{T} \mathbb{1}\{\max_{\ell_{\alpha_j} \leq s_j \leq t} \bar{X}_{s_j}(\alpha_j) + c_{s_j}(\alpha_j) \geq \min_{1 \leq s \leq t} \bar{X}_s(\alpha^*) + c_s(\alpha^*)\}$$

$$\leq \ell_{\alpha_j} + \sum_{t=1}^{T} \sum_{s=1}^{t} \sum_{s_j = \ell_{\alpha_j}}^{t} \mathbb{1}\{\bar{X}_{s_j}(\alpha_j) + c_{s_j}(\alpha_j) \geq \bar{X}_s(\alpha^*) + c_j(\alpha^*)\}$$

(20)

Following the same argument as in Auer et al. [2002a], choosing $\ell_{\alpha_j} = \frac{4\log(\sqrt{2}T/\delta)}{(\Delta'_{\alpha_j})^2}$, we have

$$\mathbb{P}(\bar{X}_{s_j}(\alpha_j) + c_{s_j}(\alpha_j) \geq \bar{X}_s(\alpha^*) + c_j(\alpha^*)) \leq \mathbb{P}(\bar{X}_s(\alpha^*) \leq \mu(\alpha^*) - c_j(\alpha^*)) + \mathbb{P}(\bar{X}_{s_j}(\alpha_j) \geq \mu(\alpha_j) + c_{s_j}(\alpha_j))$$

From Lemma A.1,

$$\mathbb{P}(\bar{X}_s(\alpha^*) \leq \mu(\alpha^*) - c_j(\alpha^*)) \leq \frac{\delta^2}{2T^2}$$

Hence,

$$\sum_{t=1}^{T} \mathbb{P}(j_t = \alpha_j) \leq \frac{4\log(\sqrt{2}T/\delta)}{(\Delta'_{\alpha_j})^2} + T\delta^2.$$

Plug into Equation (18), we can get

$$\sum_{t=1}^{T} \sum_{i \notin N_{\alpha^*}} \Delta_i \mathbb{P}(i_t = i) \leq \sum_{j=1}^{|\mathcal{I}|} \frac{(\Delta'_{\alpha_j} + 2\epsilon)4\log(\sqrt{2}T/\delta)}{(\Delta'_{\alpha_j})^2} + \Delta_{max}T\delta^2$$

$$= \sum_{j=1}^{|\mathcal{I}|} \left(\frac{1}{\Delta'_{\alpha_j}} + \frac{2\epsilon}{(\Delta'_{\alpha_j})^2}\right)4\log(\sqrt{2}T/\delta) + \sum_{j=1}^{|\mathcal{I}|} \Delta_{max}T\delta^2$$

(21)

$$\leq \frac{4\log(\sqrt{2}T/\delta)}{\epsilon}\left(\log(\alpha(G)) + \frac{\pi^2}{3}\right) + \alpha(G)\Delta_{max}T\delta^2.$$

Now we focus on the second part in Equation (18).

For any $i \in N_{\alpha^*}$, we have $\Delta_i \leq 2\epsilon$. This means the gap between suboptimal and optimal arms is bounded. Therefore, this part can be seen as using UCB-N Lykouris et al. [2020] on the graph formed by $N_{\alpha^*}$. We can directly use their results by adjusting some constant factors. Following Theorem 6 in Lykouris et al. [2020], this part has a regret upper bound as

$$16 \cdot \log(\sqrt{2}T/\delta)\log(T) \max_{I \in \mathcal{I}(N_{\alpha^*})} \sum_{i \in I\setminus\{i^*\}} \frac{1}{\Delta_i} + 2\epsilon T\delta^2 + 1 + 2\epsilon.$$

(22)

Let $\delta = \frac{1}{T}$. Combining Equation (21) and Equation (22) and using Proposition 4.1 that $\alpha(G) \leq 2\gamma(G)$, we have

$$R_T^\pi \leq \frac{4\log(\sqrt{2}T/\delta)}{\epsilon}\left(\log(\alpha(G)) + \frac{\pi^2}{3}\right) + 16 \cdot \log(\sqrt{2}T/\delta)\log(T) \max_{I \in \mathcal{I}(N_{\alpha^*})} \sum_{i \in I\setminus\{i^*\}} \frac{1}{\Delta_i} + T\delta^2(\alpha(G)\Delta_{max} + 2\epsilon) + 1 + 2\epsilon$$

$$\leq \frac{8\log(\sqrt{2}T)}{\epsilon}\left(\log(2\gamma(G)) + \frac{\pi^2}{3}\right) + 32 \cdot \log(\sqrt{2}T)\log(T) \max_{I \in \mathcal{I}(i^*)} \sum_{i \in I\setminus\{i^*\}} \frac{1}{\Delta_i} + \Delta_{max} + 4\epsilon + 1.$$

(23)

## D PROOFS OF THEOREM 4.6

We just need to discuss $\Delta_i$ in intervals

$$[0, \epsilon), [\epsilon, 2\epsilon), ..., [k\epsilon, (k+1)\epsilon), ...$$

The regret for $\Delta_i$ in $[\epsilon, 2\epsilon), ..., [k\epsilon, (k+1)\epsilon), ...$ can be bounded by the same method used in the proof of Theorem 4.3. We can calculate that the regret of this part has the same form as $O(\frac{\log(\sqrt{2}T)}{\epsilon})$.

Recall that $G_\epsilon$ denote the subgraph with arms $\{i \in V : \mu(i^*) - \mu(i) < \epsilon\}$. The set of all independent dominating sets of graph $G_\epsilon$ is denoted as $\mathcal{I}(G_\epsilon)$. The regret for $\Delta_i$ in $[0, \epsilon)$ can be bounded as

$$32(\log(\sqrt{2}T))^2 \max_{I \in \mathcal{I}(G_\epsilon)} \sum_{i \in I \setminus \{i^*\}} \frac{1}{\Delta_i}.$$

Due to the similarity assumption, $G_\epsilon$ is a complete graph. Therefore,

$$\max_{I \in \mathcal{I}(G_\epsilon)} \sum_{i \in I \setminus \{i^*\}} \frac{1}{\Delta_i} \leq \frac{1}{\Delta_{min}}.$$

## E PROOFS OF THEOREM 5.2

Recall that $\mathcal{I}_t$ denotes the independent set at time $t$ and $\alpha_t^* \in \mathcal{I}_t$ denotes the arm that include the optimal arm $i_t^*$. We have $|\mathcal{I}_T| \leq \alpha(G_T^{\mathcal{P}})$. Let $\mathcal{F}$ denote the sequences generated from $\mathcal{P}$ with length $T$, thus $\mathcal{F}$ is a random variable.

Since the optimal arm may change over time, this leads to a time-varying $\Delta_i$. We denote the new gap as $\Delta_t(i)$. Therefore, the analysis method in Theorem 4.3 is no longer applicable here. The regret can also be divided into two parts:

$$\mathbb{E}\left[\sum_{t=1}^{T} \sum_{i \in K(t)} \Delta_t(i) \mathbb{1}\{i_t = i\}\right] = \underbrace{\mathbb{E}\left[\sum_{t=1}^{T} \sum_{i \notin N_{\alpha_t^*}} \Delta_t(i) \mathbb{1}\{i_t = i\}\right]}_{(A)} + \underbrace{\mathbb{E}\left[\sum_{t=1}^{T} \sum_{i \in N_{\alpha_t^*}} \Delta_i \mathbb{1}\{i_t = i\}\right]}_{(B)} \tag{24}$$

We focus on $(A)$ first.

$$
\begin{aligned}
(A) &= \mathbb{E}_{\mathcal{F}}\left[\mathbb{E}\left[\sum_{t=1}^{T} \sum_{i \notin N_{\alpha_t^*}} \Delta_t(i) \mathbb{1}\{i_t = i\} \Big| \mathcal{F}\right]\right] \\
&= \mathbb{E}_{\mathcal{F}}\left[\mathbb{E}\left[\sum_{t=1}^{T} \Delta_t(i) \mathbb{1}\{i_t = i, j_t \neq \alpha_t^*\} \Big| \mathcal{F}\right]\right] \\
&\leq \mathbb{E}_{\mathcal{F}}\left[\mathbb{E}\left[\sum_{t=1}^{T} \Delta_{max}^T \mathbb{1}\{j_t \neq \alpha_t^*\} \Big| \mathcal{F}\right]\right]
\end{aligned}
\tag{25}
$$

Given a fixed $\mathcal{F}$, $\mathcal{I}_T$ is deterministic. Since the gap between optimal and suboptimal arms may be varying over time, we define

$$\Delta_{\alpha_j}'' = \min_{t \in [T]} \{\mu(\alpha_t^*) - \mu(\alpha_j) : \alpha_j \in \mathcal{I}_T \text{ and } \mu(\alpha_t^*) - \mu(\alpha_j) > 0\}$$

denote the minimum gap when $\alpha_j \in \mathcal{I}_T$ is not the optimal selection that include the optimal arm. Then $\Delta_{\alpha_j}'' \geq \epsilon$.

Following the proofs of Theorem 4.3, for any $\alpha_j \in \mathcal{I}_T \neq \alpha_t^*$, the probability of the algorithm selecting it will be less than $\delta^2$ after it has been selected $\frac{4\log(\sqrt{2}T/\delta)}{\epsilon^2}$ times. Therefore, the inner expectation of Equation (25) is bounded as

$$|\mathcal{I}_T| \Delta_{max}^T \left(\frac{4\log(\sqrt{2}T/\delta)}{\epsilon^2} + T\delta^2\right) \tag{26}$$

The inner expectation of Equation (25) also has a native bound $T\Delta_{max}^T$.

Plugging into $(A)$ and using Lemma A.3, we get

$$(A) \leq \mathbb{E}_{\mathcal{F}} \left[ \min\{|\mathcal{I}_T| \Delta_{max}^T (\frac{4 \log(\sqrt{2}T/\delta)}{\epsilon^2} + T\delta^2), T\Delta_{max}^T \right]$$

$$\leq \sum_{k=1}^{T} \mathbb{P}(\alpha(G_T^{\mathcal{P}}) = k) \min\{k\Delta_{max}^T (\frac{4 \log(\sqrt{2}T/\delta)}{\epsilon^2} + T\delta^2), T\Delta_{max}^T\}$$

$$= \mathbb{P}(\alpha(G_T^{\mathcal{P}}) \leq 5 \max\{\log_b T, 1\}) 5 \max\{\log_b T, 1\} \Delta_{max}^T (\frac{4 \log(\sqrt{2}T/\delta)}{\epsilon^2} + T\delta^2) + \mathbb{P}(\alpha(G_T^{\mathcal{P}}) > 5 \max\{\log_b T, 1\}) T\Delta_{max}^T$$

$$\leq 5 \max\{\log_b T, 1\} \Delta_{max}^T (\frac{4 \log(\sqrt{2}T/\delta)}{\epsilon^2} + T\delta^2) + \Delta_{max}^T. \tag{27}$$

Now we consider the part $(B)$.

$$(B) = \mathbb{E}_{\mathcal{F}} \left[ \mathbb{E} \left[ \sum_{t=1}^{T} \sum_{i \in N_{\alpha_t^*}} \Delta_i \mathbb{1}\{i_t = i\} \Big| \mathcal{F} \right] \right] \tag{28}$$

Recall that $\mathcal{A} = \{a_t : t \in [T], \mu(a_t) \in N_{\alpha_t^*}\}$, $M = \sum_{t=1}^{T} \mathbb{P}\{|\mu(a_t) - \mu(i_t^*)| \leq 2\epsilon\}$. Then

$$|\mathcal{A}| \leq M' = \sum_{t=1}^{T} \mathbb{1}\{|\mu(a_t) - \mu(i_t^*)| \leq 2\epsilon\}.$$

Since $\mu(a_t) \sim \mathcal{P}$ is independent of each other, the event $\mathbb{1}\{|\mu(a_t) - \mu(i_t^*)| \leq 2\epsilon\}$ is also mutually independent. Using Lemma A.4,

$$\mathbb{P}(M' \geq 3M) \leq e^{-M}. \tag{29}$$

Given a fixed instance $\mathcal{F}$, we divide the rounds into $L'$ parts: $(t_0 = 1, t_{L'+1} = T)$

$$[1, t_1], (t_1, t_2], ..., (t_{L'}, T].$$

This partition satisfies $\forall t \in (t_j, t_{j+1})$, $i_t^*$ is stationary. The $\alpha_t^*$ is also stationary, $\forall t \in (t_j, t_{j+1})$, let $\alpha_t^* = \alpha_j$.

Let's focus on the intervals $(t_j, t_{j+1}]$, the analysis for other intervals is similar.

The best case is that all arms in $N_{\alpha_j}$ are arrived at the beginning. In this case, the regret for this part is equivalent to the regret of using the UCB algorithm on the subgraph formed by $N_{\alpha_j}$ for $t_{j+1} - t_j$ rounds. The independence number of the subgraph formed by $N_{\alpha_j}$ is 2, which leads to a regret upper bound independent of the number of arms arriving. However, we are primarily concerned with the worst case. The worst case is that the algorithm cannot benefit from the graph feedback at all. That is, the algorithm spends $O(\frac{\log(T)}{(\Delta_1)^2})$ rounds distinguishing the optimal arm from the first arriving suboptimal arm 1. After this process, the second suboptimal arm 2 arrives, and again $O(\frac{\log(T)}{(\Delta_2)^2})$ rounds are spent distinguishing the optimal arm from this arm...

Let $V_j$ denote the arms fall into $N_{\alpha_j}$ at the rounds $(t_j, t_{j+1}]$. If $i \in V_j$ has not been arrived at round $t$, $\mathbb{P}(i_t = i) = 0$.

Following the same argument as the proofs of Theorem 4.3, the inner expectation in Equation (28) can be bounded as

$$\sum_{j=0}^{L'}\sum_{i\in V_j}\sum_{t=t_j}^{t_{j+1}}\Delta_i\mathbb{P}(i_t=i,j_t=\alpha_j)=\sum_{j=0}^{L'}\sum_{i\in V_j,\Delta_i<\Delta}\sum_{t=t_j}^{t_{j+1}}\Delta_i\mathbb{P}(i_t=i,j_t=\alpha_j)+\sum_{j=0}^{L'}\sum_{i\in V_j,\Delta_i\geq\Delta}\sum_{t=t_j}^{t_{j+1}}\Delta_i\mathbb{P}(i_t=i,j_t=\alpha_j)$$

$$\leq\sum_{j=0}^{L'}(t_{j+1}-t_j)\Delta+\sum_{j=0}^{L'}\sum_{i\in V_j,\Delta_i\geq\Delta}\left(\Delta_i(t_{j+1}-t_j)\delta^2+\frac{4\log(\sqrt{2}T/\delta)}{\Delta_i}\right)$$

$$\leq T\Delta+\sum_{j=0}^{L'}(\frac{4|V_j|\log(\sqrt{2}T/\delta)}{\Delta}+M'2\epsilon(t_{j+1}-t_j)\delta^2)$$

$$\overset{(a)}{\leq} T\Delta+\frac{4M'\log(\sqrt{2}T/\delta)}{\Delta}+2TM'\epsilon\delta^2$$

$$\overset{(b)}{\leq} 4\sqrt{TM'\log(\sqrt{2}T/\delta)}+2\epsilon,$$

(30)

where $(a)$ comes from the fact that $\sum_j|N_{\alpha_j}|\leq M'\leq T$ and $\delta=\frac{1}{T}$, $(b)$ follows from $\Delta=\sqrt{\frac{4M'\log(\sqrt{2}T/\delta)}{T}}$.

Similar to the approach of bounding $(A)$, the inner expectation in Equation (28) also has a native bound $2T\epsilon$.

Substituting into Equation (28), we get

$$(B)=\mathbb{E}_{\mathcal{F}}\left[4\sqrt{TM'\log(\sqrt{2}T/\delta)}+2\epsilon\right]$$

$$\leq\mathbb{P}(M'\leq 3M)(4\sqrt{3TM\log(\sqrt{2}T/\delta)}+2\epsilon)+2T\epsilon\mathbb{P}(M'>3M)$$

(31)

$$\leq 4\sqrt{3TM\log(\sqrt{2}T/\delta)}+2\epsilon+2T\epsilon e^{-M}.$$

From Equation (27) and Equation (31), we get the total regret

$$R_T^\pi\leq 5\max\{\log_b T,1\}\Delta_{max}^T(\frac{4\log(\sqrt{2}T/\delta)}{\epsilon^2}+T\delta^2)+\Delta_{max}^T+4\sqrt{3TM\log(\sqrt{2}T/\delta)}+2\epsilon+2T\epsilon e^{-M}.$$

# F   PROOFS OF COROLLARY 5.3

First, we calculate an $M$ that is independent of the distribution $\mathcal{P}$. Given $X_1,X_2,...,X_T$ as independent random variables from $\mathcal{P}$. Let

$$M=\sum_{t=1}^{T}\mathbb{P}(|X_t-\max_{i=1,...,t}X_i|<2\epsilon).$$

Denote $F(x)=\mathbb{P}(X<x)$, $M_t=\max_{i\leq t}X_i$. Then

$$\mathbb{P}(|X_t-M_t|<2\epsilon|M_t=x)=F(x+2\epsilon)-F(x-2\epsilon)$$

and

$$\mathbb{P}(|X_t-M_t|<2\epsilon)=t\int_{\mathcal{D}}(F(x))^{t-1}(F(x+2\epsilon)-F(x-2\epsilon))F'(x)dx,$$

where $\mathcal{D}$ is the support set of $\mathcal{P}$. Since

$$\sum_{r=1}^{R}rx^{r-1}=\frac{d}{dx}\frac{1-x^{R+1}}{1-x}=\frac{1-(R+1)x^R+Rx^{R+1}}{(1-x)^2}.$$

We get

$$M=\int_{\mathcal{D}}\frac{1-(T+1)(F(x))^T+T(F(x))^{T+1}}{(1-F(x))^2}(F(x+2\epsilon)-F(x-2\epsilon))F'(x)dx.$$

(32)

We need to estimate the upper and lower bounds of $M$ separately. For Gaussian distribution $\mathcal{N}(0, 1)$, $F(x) = \Phi(x)$, $F'(x) = \phi(x)$. We first proof $M = O(\log(T)e^{2\epsilon\sqrt{2\log(T)}})$.

Denote the integrand function as $H(x)$. Let $m = \sqrt{2\log(T)} + \epsilon$,

$$M = \int_{-\infty}^{m} H(x)dx + \int_{m}^{+\infty} H(x)dx \tag{33}$$

First, we have

$$\forall x \in \mathbb{R}, \frac{1 - (T+1)(F(x))^T + T(F(x))^{T+1}}{(1 - F(x))^2} = \sum_{t=1}^{T} t(F(x))^{t-1} \le \frac{T(T+1)}{2} \le T^2.$$

$$\Phi(x + 2\epsilon) - \Phi(x - 2\epsilon) \le 4\epsilon\phi(x - 2\epsilon).$$

And

$$(F(x + 2\epsilon) - F(x - 2\epsilon))F'(x) \le 4\epsilon\phi(x - 2\epsilon)\phi(x) \le \frac{2\epsilon e^{-\epsilon^2}}{\pi}e^{-(x-\epsilon)^2} \tag{34}$$

Then,

$$\int_{m}^{+\infty} H(x)dx \le T^2 \frac{2\epsilon e^{-\epsilon^2}}{\pi} \int_{m}^{\infty} e^{-(x-\epsilon)^2}dx$$

$$= T^2 \frac{2\epsilon e^{-\epsilon^2}}{\sqrt{\pi}}\Phi(\sqrt{2}(m - \epsilon))$$

$$\overset{(a)}{\le} T^2 \frac{2\epsilon e^{-\epsilon^2}}{\sqrt{\pi}} \frac{1}{\sqrt{2}(m - \epsilon) + \sqrt{2(m - \epsilon)^2 + 4}}e^{-(m-\epsilon)^2} \tag{35}$$

$$\le \frac{2\epsilon e^{-\epsilon^2}}{\sqrt{\pi}},$$

where $(a)$ use Lemma A.5.

Now we calculate the second term.

$$\int_{-\infty}^{m} H(x)dx = \int_{-\infty}^{1} H(x)dx + \int_{1}^{m} H(x)dx \le \frac{\Phi(1)}{(1 - \Phi(1))^2} + \int_{1}^{m} H(x)dx. \tag{36}$$

We only need to bound the integral within $(1, m)$.

$$\int_{1}^{m} H(x)dx \le \int_{1}^{m} \frac{(\Phi(x + 2\epsilon) - \Phi(x - 2\epsilon))\phi(x)}{(1 - \Phi(x))^2}dx$$

$$\le \frac{2\epsilon e^{-\epsilon^2}}{\pi} \int_{1}^{m} \frac{e^{-(x-\epsilon)^2}}{(1 - \Phi(x))^2}dx$$

$$\overset{(b)}{\le} 4\epsilon e^{-\epsilon^2} \int_{1}^{m} (\frac{1}{x} + x)^2 e^{x^2}e^{-(x-\epsilon)^2}dx \tag{37}$$

$$= 4\epsilon e^{-2\epsilon^2} \int_{1}^{m} (\frac{1}{x} + x)^2 e^{2x\epsilon}dx$$

$$\le 4m^2 e^{2m\epsilon}e^{-2\epsilon^2}$$

$$\le 8\log(T)e^{2\epsilon\sqrt{2\log(T)}},$$

where $(b)$ uses Lemma A.5.

Therefore,

$$M \le \frac{2\epsilon e^{-\epsilon^2}}{\sqrt{\pi}} + \frac{\Phi(1)}{(1 - \Phi(1))^2} + 8\log(T)e^{2\epsilon\sqrt{2\log(T)}}.$$

Then we derive a lower bound for $M$.

Since $\phi(t)$ is convex function in $[1, +\infty)$, we have

$$\phi(t) \geq \phi(\frac{a+b}{2}) + \phi'(\frac{a+b}{2})(t - \frac{a+b}{2}), \forall t \in [a, b], a \geq 1.$$

Then when $x - 2\epsilon \geq 1$,

$$\Phi(x + 2\epsilon) - \Phi(x - 2\epsilon) = \int_{x-2\epsilon}^{x+2\epsilon} \phi(t)dt \geq 4\epsilon\phi(x).$$

Hence,

$$(\Phi(x + 2\epsilon) - \Phi(x - 2\epsilon))\phi(x) \geq 4\epsilon(\phi(x))^2.$$

Substituting into Equation (32),

$$
\begin{aligned}
M &\geq \int_{1+2\epsilon}^{\sqrt{\log(T)}} \frac{1 - (T+1)(\Phi(x))^T + T(\Phi(x))^{T+1}}{(1 - \Phi(x))^2} 4\epsilon(\phi(x))^2 dx \\
&\geq \frac{8\epsilon}{\pi} \int_{1+2\epsilon}^{\sqrt{\log(T)}} (x^2 + 1)(1 - (T+1)\left(1 - \frac{1}{\sqrt{2\pi}} \frac{x}{1+x^2} e^{-\frac{x^2}{2}}\right)^T + T(\Phi(x))^{T+1})dx \\
&\geq \frac{8\epsilon}{\pi} \int_{1+2\epsilon}^{\sqrt{\log(T)}} (x^2 + 1)(1 - (T+1)\left(1 - \frac{1}{\sqrt{2\pi}} \frac{x}{1+x^2} e^{-\frac{x^2}{2}}\right)^T)dx.
\end{aligned}
\tag{38}
$$

The function $h(x) = 1 - \frac{1}{\sqrt{2\pi}} \frac{x}{1+x^2} e^{-\frac{x^2}{2}}$ is increasing on the interval $(1, +\infty)$. We have

$$
\begin{aligned}
(1 - \frac{1}{\sqrt{2\pi}} \frac{x}{1 + x^2} e^{-\frac{x^2}{2}})^T &\leq (1 - \frac{1}{\sqrt{2\pi}} \frac{\sqrt{\log(T)}}{1 + \log(T)} \frac{1}{\sqrt{T}})^T \\
&\leq e^{-\sqrt{\frac{T}{4\pi \log(T)}}}
\end{aligned}
\tag{39}
$$

Observe that for large $T$ ( $T \geq e^{11}$), $e^{-\sqrt{\frac{T}{4\pi \log(T)}}} \leq \frac{1}{T^2}$. Therefore, for any $T \geq e^{11}$,

$$M \geq \frac{8\epsilon}{\pi} \int_{1+2\epsilon}^{\sqrt{\log(T)}} (x^2 + 1)(1 - \frac{T+1}{T^2})dx \geq \frac{8\epsilon}{\pi} \int_{1+2\epsilon}^{\sqrt{\log(T)}} x^2 dx.$$

We have

$$M \geq \frac{8\epsilon}{3\pi} \log(T)\sqrt{\log(T)} - \frac{8\epsilon(1 + 2\epsilon)^2}{3\pi}.$$

## G  PROOFS OF THEOREM 5.5

Similar to the proofs of Theorem 5.2, the regret can also be divided into two parts:

$$\mathbb{E}\left[\sum_{t=1}^{T} \sum_{i \in K(t)} \Delta_t(i)\mathbb{1}\{i_t = i\}\right] = \underbrace{\mathbb{E}\left[\sum_{t=1}^{T} \sum_{i \notin N_{\alpha_t^*}} \Delta_t(i)\mathbb{1}\{i_t = i\}\right]}_{(A)} + \underbrace{\mathbb{E}\left[\sum_{t=1}^{T} \sum_{i \in N_{\alpha_t^*}} \Delta_i \mathbb{1}\{i_t = i\}\right]}_{(B)} \tag{40}$$

Part $(A)$ is exactly the same as the analysis in the corresponding part of Theorem 5.2. We only focus part $(B)$.

Let

$$L' = \sum_{t=1}^{T} \mathbb{1}\{\mu(a_t) > \mu(i_t^*)\}$$

denote the number of times the optimal arms changes. Then

$$\mathbb{E}[L'] = L = \sum_{t=1}^{T} \mathbb{P}\{\mu(a_t) > \mu(i_t^*)\}.$$

Using Lemma A.4, we can get

$$\mathbb{P}(L' \geq 3L) \leq e^{-L}. \tag{41}$$

Since $\mu(a_t) \sim \mathcal{P}$ is independent of each other, each $\mu(a_t)$ is equally likely to be the largest one, i.e., $\mathbb{P}(\mu(a_t) > \mu(i_t^*)) = \frac{1}{t}$. Or we can obtain this result through Equation (32). We have

$$L = \sum_{t=1}^{T} \frac{1}{t} = \log(T) + c + o(1), \tag{42}$$

where $c \approx 0.577$ is the Euler constant. Then $\log(T) \leq L \leq \log(T) + 1$.

Given a fixed instance $\mathcal{F}$, we also divide the rounds into $L'$ parts: $(t_0 = 1, t_{L'+1} = T)$

$$[1, t_1], (t_1, t_2], ..., (t_{L'}, T].$$

This partition satisfies $\forall t \in (t_j, t_{j+1})$, $i_t^*$ is stationary. The $\alpha_t^*$ is also stationary, $\forall t \in (t_j, t_{j+1})$, let $\alpha_t^* = \alpha_j$.

Let's focus on the intervals $(t_j, t_{j+1}]$, the analysis for other intervals is similar. All arms falling into $N_{\alpha_j}$ at rounds $(t_j, t_{j+1})$ are denoted by $V_j$. The arms in $V_j$ can be divided into two parts: $E_1 = \{i \in V_j, \mu(i) < \mu(\alpha_j)\}$ and $E_2 = \{i \in V_j, \mu(i) \geq \mu(\alpha_j)\}$. If $i \in V_j$ has not been arrived at round $t$, $\mathbb{1}\{i_t = i\} = 0$. Then we have

$$\sum_{i \in V_j} \sum_{t=t_j}^{t_{j+1}} \mathbb{1}\{i_t = i\} = \underbrace{\sum_{i \in E_1} \sum_{t=t_j}^{t_{j+1}} \mathbb{1}\{i_t = i\}}_{(C)} + \underbrace{\sum_{i \in E_2} \sum_{t=t_j}^{t_{j+1}} \mathbb{1}\{i_t = i\}}_{(D)} \tag{43}$$

From the way our algorithm constructs independent sets, it can be inferred that all arms in $V_j$ are connected to $\alpha_t^* = \alpha_j$, and the distances are all less than $\epsilon$. Hence, both $E_1$ and $E_2$ form a clique.

Note that selecting any arm in $E_1$ will result in the observation of $\alpha_j$. We have

$$
\begin{aligned}
(C) = \sum_{i \in E_1} \sum_{t=t_j}^{t_{j+1}} \mathbb{1}\{i_t = i\} &\leq \ell_{\alpha_j} + \sum_{i \in E_1} \sum_{t=t_j}^{t_{j+1}} \mathbb{1}\{i_t = i, O_t(\alpha_j) > \ell_{\alpha_j}\} \\
&\leq \ell_{\alpha_j} + \sum_{i \in E_1} \sum_{t=t_j}^{t_{j+1}} \mathbb{1}\{\bar{\mu}_t(i) - c_t(i) > \bar{\mu}_t(\alpha_j) - c_t(\alpha_j), O_t(\alpha_j) \geq \ell_{\alpha_j}\} \\
&\leq \ell_{\alpha_j} + \sum_{i \in E_1} \sum_{t=t_j}^{t_{j+1}} \mathbb{1}\{\max_{1 \leq s_i \leq t} \bar{X}_{s_i}(i) - c_{s_i}(i) > \min_{\ell_{\alpha_j} \leq s \leq t} \bar{X}_s(\alpha_j) - c_s(\alpha_j)\} \\
&\leq \ell_{\alpha_j} + \sum_{i \in E_1} \sum_{t=t_j}^{t_{j+1}} \sum_{s_i=1}^{t} \sum_{s=\ell_{\alpha_j}}^{t} \mathbb{1}\{\bar{X}_{s_i}(i) - c_{s_i}(i) > \bar{X}_s(\alpha_j) - c_s(\alpha_j)\}
\end{aligned} \tag{44}
$$

Let $\ell(\alpha_j) = \frac{4 \log(\sqrt{2}T/\delta)}{(\Delta_{min}^T)^2}$. If $s \geq \ell(\alpha_j)$, the event $\{\mu(\alpha_j) - \mu(i) \leq 2c_{s_i}(i)\}$ never occurs. Then

$$\{\bar{X}_{s_i}(i) - c_{s_i}(i) > \bar{X}_s(\alpha_j) - c_s(\alpha_j)\} \subset \{\bar{X}_{s_i}(i) > \mu(i) + c_{s_i}(i)\} \bigcup \{\bar{X}_s(\alpha_j) < \mu(\alpha_j) - c_s(\alpha_j)\}.$$

From Lemma A.1, we have

$$\mathbb{P}(\bar{X}_{s_i}(i) - c_{s_i}(i) > \bar{X}_s(\alpha_j) - c_s(\alpha_j)) \leq \frac{\delta^2}{T^2}.$$

The regret incurred by $E_1$ in $(t_j, t_{j+1}]$ is at most

$$\frac{8\epsilon \log(\sqrt{2}T/\delta)}{(\Delta_{min}^T)^2} + 2\epsilon(t_{j+1} - t_j)|E_1|\delta^2.$$

Using the same method, we can get the regret incurred by $E_2$ in $(t_j, t_{j+1}]$ is bounded as

$$\frac{8\epsilon \log(\sqrt{2}T/\delta)}{(\Delta_{min}^T)^2} + 2\epsilon(t_{j+1} - t_j)|E_2|\delta^2.$$

Therefore, choosing $\delta = \frac{1}{T}$, $(B)$ with the fixed $\mathcal{F}$ is bounded as

$$\frac{16L'\epsilon \log(\sqrt{2}T/\delta)}{(\Delta_{min}^T)^2} + 2\epsilon. \tag{45}$$

From Equation (41) and Equation (42), we have

$$(B) \leq \frac{48L\epsilon \log(\sqrt{2}T/\delta)}{(\Delta_{min}^T)^2} + 2\epsilon + 2\epsilon T e^{-L} \leq \frac{48L\epsilon \log(\sqrt{2}T/\delta)}{(\Delta_{min}^T)^2} + 4\epsilon$$

Therefore, we get the total regret

$$R_T^\pi \leq 5 \log_b T \Delta_{max} \left( \frac{4 \log(\sqrt{2}T/\delta)}{\epsilon^2} + T\delta^2 \right) + \Delta_{max} + \frac{48L\epsilon \log(\sqrt{2}T/\delta)}{(\Delta_{min}^T)^2} + 4\epsilon.$$