# OpenReview forum: "Graph Feedback Bandits with Similar Arms"
_auai.org/UAI/2024/Conference — UAI 2024 spotlight_

### Official Review · Reviewer_D7Qb · 2024-02-29

**Q2-1 Originality-Novelty:** 2
**Q2-2 Correctness-Technical Quality:** 2
**Q2-5 Clarity Of Writing:** 3

**Q1 Summary And Contributions:**

This draft tries to study Graph Feedback stochastic Multi-armed Bandits problems with Similar Arms that are naturally linked with clustering, where two UCB based algorithms are proposed

**Q2-3 Extent To Which Claims Are Supported By Evidence:**

2: Fair: the main claims are somewhat supported by evidence (but the experimental evaluation may be weak, or does not match entirely with the claims, important baselines may be missing, proofs contain important ideas but lack rigor, algorithmic details are only discussed superficially, references are imprecise, assumptions are not sufficiently motivated or explicated, etc.).

**Q2-4 Reproducibility:**

3: Good: key resources (e.g. proofs, code, data) are available and key details (e.g. proofs, experimental setup) are sufficiently well-described for competent researchers to confidently reproduce the main results.

**Q3 Main Strengths:**

1. this draft is easy to understand and follow
2. try to study similar arms in a way similar to clustering is important, especially when the number of arms is increasing

**Q4 Main Weakness:**

1. your empirical performance is not impressive, and how this proposal is useful in practice or industry is unclear, which should be clearly articulated
2. in your experiments, the data scale is a bit small and you don't have large-scale and real-world or production data based experimental results to support your claims which is one of the main drawbacks of this work

**Q5 Detailed Comments To The Authors:**

This manuscript studies graph feedback bandits models for similar arms and extended to ballooning setting, with the potential to incorporate similar techniques on bandits, there are related state-of-the-art you may want to compare: The Art of Clustering Bandits, Fast Distributed Bandits for Online Recommendation Systems

**Q9 Complying With Reviewing Instructions:**

Yes

---

> ### Author Rebuttal · Authors · 2024-04-05
>
> Thank you for your review.  I'm glad to see that you mentioned clustering bandits. I believe there are interesting connections.
>
> Our contribution is divided into two parts. The first part mainly focuses on theoretical aspects in stationary scenarios. The second part, the ballooning setting, is particularly useful in real-world applications, such as  question-and-answer (Q&A) platforms, product reviews and hotel reviews. In my responses to Reviewers 84Wn and mjGw, I provided more detailed explanations of the motivation. We will incorporate these into the main text.
>
> I agree with you about the data scale. Using $K=10000$ might be a bit small compared to real-world applications. We will conduct experiments on large datasets as soon as possible. I believe the experimental results will not differ significantly from those presented in the paper.

---

### Official Review · Reviewer_nBAe · 2024-03-20

**Q2-1 Originality-Novelty:** 3
**Q2-2 Correctness-Technical Quality:** 3
**Q2-5 Clarity Of Writing:** 3

**Q1 Summary And Contributions:**

The article investigates the setting of Graph Feedback Bandits, building upon the Multi-Armed Bandit (MAB) framework. Specifically, when pulling an arm $i$, observations are not limited to arm $i$ alone; information about other arms $j$ that are $\epsilon$-similar to arm $i$ is also obtained. The paper explores two scenarios: one in which the set of arms remains unchanged, and another described as the ballooning bandit, where the number of arms increases over time. The aim of this study is to optimize the expected regret. The paper establishes a lower bound on the regret for the problem of graph feedback bandits and introduces two strategies, D-UCB and C-USB, to address this challenge.

**Q2-3 Extent To Which Claims Are Supported By Evidence:**

3: Good: the main claims are supported by convincing evidence (in the form of adequate experimental evaluation, proofs, (pseudo-)code, references, assumptions).

**Q2-4 Reproducibility:**

3: Good: key resources (e.g. proofs, code, data) are available and key details (e.g. proofs, experimental setup) are sufficiently well-described for competent researchers to confidently reproduce the main results.

**Q3 Main Strengths:**

This work considers a novel setting, proposes improved algorithms, and the theoretical foundations are solid. The paper is well-written and qualifies as a competent piece of research.

**Q4 Main Weakness:**

I believe there are some aspects of the paper that could benefit from a more intuitive physical explanation. For instance, a key physical quantity, the dominant number $\gamma(G)$, is introduced in the introduction and used as a parameter characterizing the problem in the problem-dependent bound. However, its meaning and significance seem to be inadequately explained, with only a definition provided. There is little discussion on why it represents the difficulty of the problem or how it affects the algorithm's performance.

**Q5 Detailed Comments To The Authors:**

In the conventional Multi-Armed Bandit (MAB) setting, the UCB algorithm can achieve a regret bound of $O\left(\sum_{i \in I \setminus \{i^*\}} \frac{\ln T}{\Delta_i}\right)$, but when looking at the regret bound in Theorem 4.3, it doesn't seem to improve. This raises curiosity because the graph should provide additional information, yet this advantage doesn't appear to reflect in the results.

**Q9 Complying With Reviewing Instructions:**

Yes

---

> ### Author Rebuttal · Authors · 2024-04-05
>
> Thank you for your review; it has inspired me on how to improve our article. Now I would like to focus on two of your concerns.
>
> **Dominant number $\gamma(G)$:** Proposition 4.1 shows that  $\gamma(G) \leq \alpha(G) \leq 2\gamma(G)$ for our similarity feedback graph $G$.  General graphs do not have this property.  Our theoretical analysis framework is based on independent sets, and using this proposition we can immediately get an upper bound on $\gamma(G)$. For general graph feedback bandit problems,  without using the feedback graph to explicitly target exploration, it is difficult to get an upper bound on $\gamma(G)$. Therefore, $\gamma(G)$ is introduced to show that, with the help of similarity structures, our algorithm can still obtain a smaller upper bound without using graph information.
>
> **Q5:** The upper bound in Theorem 4.3 related to $\Delta_i$ is
>
> $$ \max_{I \in \mathcal{I}(i^{*} )}\sum_{i \in I \setminus i^{\ast} } \frac{1}{\Delta_i}$$
>
> From the definition of $\mathcal{I}(i^{\ast})$ (Section 4.2.1), $\forall I \in \mathcal{I}(i^{*})$, we have $|I| \leq 2$. So there are at most two terms in this sum. This is an improvement over traditional literature. In our reply to Reviewer 84Wn, we provided a more detailed explanation.

---

### Official Review · Reviewer_mjGw · 2024-03-21

**Q2-1 Originality-Novelty:** 3
**Q2-2 Correctness-Technical Quality:** 3
**Q2-5 Clarity Of Writing:** 4

**Q1 Summary And Contributions:**

Building on the traditional graph feedback bandits model,  this paper focuses on a model of bandits in which an underlying graph connects similar arms (which means rewards are within some epsilon of each other). Besides providing a lower bound for this problem, the authors also provide two algorithms. They also consider the ballooning bandits setting, in which the number of arms grows over time. In this framework, the similarity structure helps, because the learner can afford not to test an arm which is known to be similar to a suboptimal arm.

**Q2-3 Extent To Which Claims Are Supported By Evidence:**

3: Good: the main claims are supported by convincing evidence (in the form of adequate experimental evaluation, proofs, (pseudo-)code, references, assumptions).

**Q2-4 Reproducibility:**

3: Good: key resources (e.g. proofs, code, data) are available and key details (e.g. proofs, experimental setup) are sufficiently well-described for competent researchers to confidently reproduce the main results.

**Q3 Main Strengths:**

The paper is well written and well-organized.

The analysis elegantly mixes elements from graph theory and bandits.

The lower and upper bounds match up to a logarithmic term. Considering a bandits setting in which the set of arms evolves is in practice, very interesting.

**Q4 Main Weakness:**

I think that the knowledge of a similarity graph is hard to justify in practice. What happens if the given similarity graph is not completely trustworthy? What if some edges are missing or if there are too many edges?

The ballooning setting is also not motivated enough in my opinion.

**Q5 Detailed Comments To The Authors:**

Typos:

4.3, before 4.3.1 : which simply modifiES

5.1.1 It’s easY to verify

Appendix B: We first proVE

**Q9 Complying With Reviewing Instructions:**

Yes

---

> ### Author Rebuttal · Authors · 2024-04-05
>
> Thank you for your feedback. We're glad to see your interest in our work.
>
> I agree with you about the difficulty of judging similarity. In practice, we can manually detect whether there is little difference in the means of the arms that have been connected together.
>
> Regarding the case where the given similarity graph is not completely trustworthy, we have previously considered another model where the smaller the difference in the means, the greater the probability of being connected together. This model may be able to address your proposed situation and  seems to be more in line with reality. However, theoretical analysis became more difficult, so we chose the model in the present paper.  That would be an interesting line of research.
>
> In our reply to Reviewer 84Wn about **Q5-1**, we added some motivation for the "ballooning setting". It  mainly involves question-and-answer (Q&A) platforms. There are also other practical applications, such as  Amazon and Flipkart (product reviews), Tripadvisor (hotel reviews).  As time goes by, comments on products (or hotels or movies) keep appearing on the website, and the goal is to display the most useful comments on top of the product (or hotel or movie). The usefulness of the comments is evaluated based on user approval, similar to question-and-answer (Q&A) platforms.

---

### Official Review · Reviewer_PkLz · 2024-03-23

**Q2-1 Originality-Novelty:** 4
**Q2-2 Correctness-Technical Quality:** 3
**Q2-5 Clarity Of Writing:** 4

**Q10 Ethical Concerns:**

No ethical concern.

**Q1 Summary And Contributions:**

This paper investigates the MAB problem with graph feedback and assumes partial observation (neighbor of the selected arm). The author provided the D-UCB algorithm and C-UCB algorithm and analyzes the two algorithms and their variants under the standard graph feedback bandit problem and the bandit problem with an increasing number of arms.

**Q2-3 Extent To Which Claims Are Supported By Evidence:**

3: Good: the main claims are supported by convincing evidence (in the form of adequate experimental evaluation, proofs, (pseudo-)code, references, assumptions).

**Q2-4 Reproducibility:**

3: Good: key resources (e.g. proofs, code, data) are available and key details (e.g. proofs, experimental setup) are sufficiently well-described for competent researchers to confidently reproduce the main results.

**Q3 Main Strengths:**

The new settings introduced in paper may have some practical applications. It presents an algorithm with theoretical guarantees on regret and rate violation. The main idea is using the non-negative state variable $Q_i$'s to guide the algorithm towards fairness while use standard updates for the sample distributions. Additionally, empirical evaluations are provided to support the proposed approach.

**Q4 Main Weakness:**

D-UCB and C-UCB might be agnostic to graph information, but it's unclear if Double-UCB for Ballooning Setting and C-UCB for Ballooning Setting are. Should we pull arm $a_t$ around line 5 to update the neighbor set $N$ for all arms in the independent set $\mathcal{I}$? This adjustment might make the algorithm agnostic to graph information as well.

**Q5 Detailed Comments To The Authors:**

1. There are some typos in the paper:
- In all algorithms, updates for the number of observations $O$ and mean estimator $\bar\mu$ should be updated for all arms in $N_i(t)$. And the update $t = t + 1$ should not be in the algorithm.
- Theorem 4.5 has $\delta$ in the upper bound, which should be set to $1/T$.

2. Is it possible to derive a bound from Theorem 5.4 for the normal distribution? This question won't affect my rating for the paper.

3. Could the author provide some explanation as to why UCB-N-Standard performs better than UCB-N in Figure 1? Are there differences in the settings?

**Q9 Complying With Reviewing Instructions:**

Yes

---

> ### Author Rebuttal · Authors · 2024-04-05
>
> Thank you for your helpful suggestions, we will promptly address those typos. Next, I will add some clarification to your comments in **Q4** and **Q5**.
>
> **Q4:** I agree with you that D-UCB and C-UCB are not agnostic to graph information in ballooning settings.  Thanks  for proposing an improved method. In my opinion, your suggestion may be helpful. However, for the arrived arm $a_t$, the algorithm needs to check if it is in $N_{\mathcal{I}}$, which requires knowledge of the graph. So, the problem has not been completely resolved.
>
> **Q5-2:**  For the normal distribution, we can also  derive a bound from Theorem 5.4.  The key is to calculate the lower bound of $B$. Since (due to some problems with MathJax, I changed the representation of $B$),
> $$ B \geq  \mathbb{E}\Big[ \sum_{t=1}^{T} I (  \frac{\epsilon}{2}< \mu(i_t^{*})-\mu(a_t) < \frac{\epsilon}{2}   )  \Big]$$
> This lower bound is similar to the definition of $M$ (Eq 11). Using almost the same method as the proof for Corollary 5.3, we can calculate that the lower bound of $B$ for Gaussian is $\Omega(\log(T)\sqrt{\log(T)})$.
>
> In my opinion, the lower bound given by theorem 5.4 is relatively simple and may not be a strict lower bound for the algorithm. For uniform distribution and half-triangle distribution, this lower bound is sufficient to show that the D-UCB algorithm has linear regrets. For Gaussian distributions, this lower bound does not reveal more useful information.
>
> **Q5-3:** It seems that there are some typos in this question. Our results show that UCB-N performs better than UCB-N-Standard in Figure 1.
> Both cases use the same algorithm UCB, but have different experimental settings.  The purpose of this experiment is to show that the similarity structure improves the performance of the original UCB algorithm.
>
> "UCB-N:" Graph feedback **with** similarity structure
>
> "UCB-N-Standard: " Graph feedback **without** similarity structure
>
> To ensure fairness, the problem instances we use in both cases have roughly the same independence number.  We realize that this setting may not be clearly described in the main text, and we will improve the description of this experimental setting later.

---

### Official Review · Reviewer_84Wn · 2024-03-23

**Q2-1 Originality-Novelty:** 3
**Q2-2 Correctness-Technical Quality:** 3
**Q2-5 Clarity Of Writing:** 3

**Q10 Ethical Concerns:**

No.

**Q1 Summary And Contributions:**

The paper studies the stochastic multi-armed bandit problem with graph feedback. The contribution is three folds. First, they propose a new graph feedback model where an edge is formed when the means of two arms is close. Second, they propose two algorithms, D-UCB, C-UCB for this setting and provided regret bounds. Finally, they extend their algorithms to the "ballooning setting" where number of arms increases over time.

**Q2-3 Extent To Which Claims Are Supported By Evidence:**

4: Excellent: all claims are supported by very convincing evidence (in the form of comprehensive experimental evaluation, rigorous mathematical proofs, detailed (pseudo-)code, precise references, well-motivated and realistic assumptions) and the authors deliver what they promise.

**Q2-4 Reproducibility:**

3: Good: key resources (e.g. proofs, code, data) are available and key details (e.g. proofs, experimental setup) are sufficiently well-described for competent researchers to confidently reproduce the main results.

**Q3 Main Strengths:**

The paper is clearly written. Theoretical results are clearly stated and justified. The new model considered is novel to me. The new approach and its analysis seem novel to me as well.

**Q4 Main Weakness:**

See Q5.

**Q5 Detailed Comments To The Authors:**

1. I suggest adding stronger motivations for the "ballooning setting".
2. The problem dependent bounds are related to a $\Delta_{min}$ term. This term can be extremely small so the resulting regret can be very large. Compared to some classical bounds in MAB literature, I think terms like $\sum_i 1/\Delta_i$ is typical.
3. It seems that the only difference of C-UCB compared with D-UCB is using LCB in arm selection step within a similar group of arms. Intuitively I think this selection rule does not favor exploration but I'm not sure why it works.

**Q9 Complying With Reviewing Instructions:**

Yes

---

> ### Author Rebuttal · Authors · 2024-04-05
>
> Thank you very much for your review. Next, I will try to provide some explanations regarding your detailed comments in **Q5**.
>
> **Q5-1:** We should indeed add some motivation for the "ballooning setting". This setting was first proposed by  Ghalme et al.
> 2021 .
>
> A representative example is the question-and-answer (Q&A) platforms such as Reddit, Stack Overflow, Quora, Yahoo! Answers, and ResearchGate, where the goal of the platform is to discover the highest quality answers to given questions to display prominently.  The platform may model this as an MAB problem, where choosing an arm is equivalent to clicking or agreeing. Each answer post is modeled as an arm of a multi-armed bandit (MAB) instance, and rewards are allocated based on some distribution.  If a user likes the answer shown to her, then she may approve of the answer or give it a higher score.  Further, the user may also choose to post her own answer, thus increasing the number of available arms. Hence, the number of available arms (answers) monotonically increases over time.
>
> **Q5-2:** Equation 6 provides an upper bound related to $\Delta_{min}$, this is mainly for comparison with the lower bound in Theorem 4.2 .  To see more clearly the advantages of our upper bound over the classical graph-feedback MAB literature, we should focus on Theorem 4.3.
>
> The upper bound in Theorem 4.3 related to $\Delta_i$ is
>
> $$ \max_{I \in \mathcal{I}(i^{*} )}\sum_{i \in I \setminus i^{\ast} } \frac{1}{\Delta_i}$$
>
> From the definition of $\mathcal{I}(i^{\ast})$ (Section 4.2.1), $\forall I \in \mathcal{I}(i^{*})$, we have $|I| \leq 2$.
>
> Therefore, classical upper bounds like $\sum_{i}\frac{1}{\Delta_i}$ have the sum of gaps of $K$ arms or $\alpha(G)$ arms, while ours have at most two.  Although the upper bound of Theorem 4.3 has one more logarithmic term, the upper bound of our algorithm shows an advantage when  the order of the number of arms $K$ or  independent numbers $\alpha(G)$ is greater than $\log T$.
>
> **Q5-3:** Section 4.3.1 details why the C-UCB algorithm works. Intuitively (or from Eq 7-8 ), as long as the optimal arm can be observed more  than $O(\log T)$ times (ignoring terms independent of $T$), the algorithm will select the optimal arm with high probability.
>
> The similarity structure can ensure that the optimal arm can be observed enough times. Since in the graph formed by $N_{j_t}$, all arms are connected to  $j_t$. As long as the time steps are sufficiently large, arm $j_t$  will inevitably be observed more than $O(\log T)$ times, then the arms with means less than $\mu(j_t)$ will be ignored. Choosing arms  that the means between $(\mu(j_t),\mu(i^{*}))$  will always observe the optimal arm, so the optimal arm can be observed $O(\log T)$ times.
>
> I hope my reply will clarify your concerns.

---

### Meta-Review · Area_Chair_Qiyd · 2024-04-22

The paper explores the stochastic multi-armed bandit problem with graph feedback, proposing a novel model where arms with similar means are connected by edges. It introduces two algorithms, D-UCB and C-UCB, tailored for this context and establishes regret bounds for each. Additionally, the study extends these algorithms to the "ballooning setting," where the number of arms increases over time. Leveraging the similarity structure, the learner can refrain from testing arms similar to known suboptimal choices, providing a strategic advantage in the ballooning bandits scenario.

The work addresses a novel setting (especially the ballooning setting) and proposes improved algorithms with good theoretical guarantees. The paper is well-written. We are happy to accept the paper, however, the authors are strongly advised to incorporate the reviewers' suggestions: include large data experiments as pointed by Reviewer D7Qb, add a comparison with clustering bandits and clarify the practical motivations properly in the abstract and the introduction, add a clear comparison between UCB-N-Standard and UCB-N in the Expt section. Also, several typos are needed to be corrected. Please address these issues in the camera-ready version without fail.

Thanks
AC